# On the importance of discharge observation uncertainty when interpreting hydrological model performance

Jerom P.M. Aerts[1], Jannis M. Hoch[2,3], Gemma Coxon[4], Nick C. van de Giesen[1], and Rolf W. Hut[1]

[1]Department of Water Management, Civil Engineering and Geoscience, Delft University of Technology, Delft, the Netherlands
[2]Department of Physical Geography, Utrecht University, Utrecht, the Netherlands
[3]Fathom, Bristol, United Kingdom
[4]Geographical Sciences, University of Bristol, Bristol, United Kingdom

**Correspondence:** Jerom Aerts (j.p.m.aerts@tudelft.nl)

**Abstract.** For users of hydrological models, the suitability of models can depend on how well their simulated outputs align with observed discharge. This study emphasizes the crucial role of factoring in discharge observation uncertainty when assessing the performance of hydrological models. We introduce an ad-hoc approach, implemented through the eWaterCycle platform, to evaluate the significance of differences in model performance while considering the uncertainty associated with discharge observations. The analysis of the results encompasses 299 catchments from the CAMELS-GB large-sample catchment dataset, addressing 3 practical use cases for model users. These use cases involve assessing the impact of additional calibration on model performance using discharge observations, conducting conventional model comparisons, and examining how the variations in discharge simulations resulting from model structural differences compare with the uncertainties inherent in discharge observations.

Based on the 5th to 95th percentile range of observed flow, our results highlight the substantial influence of discharge observation uncertainty on interpreting model performance differences. Specifically, when comparing model performance before and after additional calibration, we find that in 98 out of 299 instances, the simulation differences fall within the bounds of discharge observation uncertainty. This underscores the inadequacy of neglecting discharge observation uncertainty during calibration and subsequent evaluation processes. Furthermore, in the model comparison use case, we identify numerous instances where observation uncertainty masks discernible differences in model performance, underscoring the necessity of accounting for this uncertainty in model selection procedures. While our assessment of model structural uncertainty generally indicates that structural differences often exceed observation uncertainty estimates, few exceptions exist. The comparison of individual conceptual hydrological models suggests no clear trends between model complexity and subsequent model simulations falling within the uncertainty bounds of discharge observations.

Based on these findings, we advocate integrating discharge observation uncertainty into the calibration process and the reporting of hydrological model performance, as has been done in this study. This integration ensures more accurate, robust, and insightful assessments of model performance, thereby improving the reliability and applicability of hydrological modeling outcomes for model users.

## 1   Introduction

Many fields in geoscience rely on uncertain data to accurately estimate states and fluxes that support decision-making. Uncertain hydrology data encompasses multiple sources, including direct measurements, proxy-based measurements, interpolation techniques, scaling processes, and data management practices (McMillan et al. (2018). A large amount of literature has been devoted to discussing the effect of data quality limitations on hydrological modeling (e.g. Yew Gan et al. (1997); Kirchner

(2006); Beven et al. (2011); Kauffeldt et al. (2013); Huang and Bardossy (2020); Beven et al. (2011); Beven and Smith (2015); Beven (2016); Beven and Lane (2022); Beven et al. (2022)). Data uncertainty can be distinguished into input data uncertainty (e.g., Kavetski et al. (2006a, b)) and evaluation data uncertainty (e.g., McMillan et al. (2010).

Input data primarily comprises meteorological variables such as precipitation and temperature. Other input data sources include static data, such as soil and topographic properties used to estimate model parameters. The inherent uncertainties in

input datasets influence the model's simulation of states and fluxes (e.g. Balin et al. (2010); Bárdossy and Das (2008); Bárdossy et al. (2022); Bárdossy and Anwar (2023); McMillan et al. (2011); Beven (2021)). The uncertainty propagation from input to model output is also closely influenced by the model structure (Butts et al., 2004; Liu and Gupta, 2007; Zhou et al., 2022; Montanari and Di Baldassarre, 2013). The effects of uncertainty propagation have been a focal point in literature, e.g., Beven (2006); Montanari and Toth (2007); Gupta and Govindaraju (2019).

Evaluation data uncertainty, the focus of this study, plays a pivotal role in determining hydrological models' potential accuracy and robustness. This is the case for model calibration, a process that involves fine-tuning model parameters to ensure that the model accurately and consistently reflects the observed historical behavior of the hydrologic system. Typically, this is based on discharge. When a model aims to replicate discharge values without including discharge observation uncertainty, the results are constrained to match a precise but potentially inaccurate representation of the hydrological response (Vrugt et al., 2005).

Consequently, accurately calibrating the model becomes more challenging due to the demand of incorporating evaluation data uncertainty into the calibration process to minimize bias in model parameters (McMillan et al., 2010).

Multiple studies have demonstrated the importance of accounting for uncertainties in discharge observations. These mainly focus on hydrological model calibration (e.g., Beven and Binley (1992); Beven and Freer (2001); Beven and Smith (2015); McMillan et al. (2018); Beven and Lane (2019); Westerberg et al. (2020, 2022); McMillan et al. (2010); Coxon et al. (2015); Liu

et al. (2009); Blazkova and Beven (2009)). In these studies multiple methodologies are used to quantify uncertainty estimates of discharge observations that are subsequently used for model calibration (overview in McMillan et al. (2012)).

Combined, all uncertainty sources (input data, evaluation data, model structure, model parameters, initial conditions) add to a concept in hydrological modeling commonly referred to as the equifinality concept (Beven and Freer (2001); Beven (2006); Montanari and Grossi (2008); Clark et al. (2008); Beven et al. (2011)). The concept is characterized by the circumstance of

various model configurations yielding similar behavioral or acceptable results. Therefore, the recommendation is to account for all uncertainty sources simultaneously. An example of a method that includes all uncertainty sources during the parameter

estimation process is the General Likelihood Uncertainty Estimation (GLUE; Beven and Freer (2001)) method. In practice, such methods are not always applied by model users, although the difficulty of implementation can be dispelled (Pappenberger and Beven (2006)).

Hydrological model evaluation by model users is often solely based on discharge observations. The inherent uncertainties in this single source of observational data might obscure the model's ability to simulate actual discharge. Therefore, omitting data uncertainty during model evaluation negatively affects the interpretation of relative model simulation differences, as these might fall within the uncertainty bounds of the observations.

Another challenging aspect of hydrological modeling is the hydrological system's large spatial and temporal variability.
The large variety in landscape and hydrological heterogeneity can be captured when evaluating or comparing hydrological models using so-called large-sample catchment hydrology datasets. These large-sample catchment datasets contain hydro-meteorological time series, catchment boundaries, and catchment attributes for a large set of catchments. The dataset is complemented with discharge observations at the catchment outlets and meteorological forcing datasets that include precipitation, temperature, and reference evaporation. The large-sample catchment datasets are collected using a consistent methodology
across all catchments.

Recent large-sample datasets follow the structure introduced by (Addor et al., 2017) in the form of the CAMELS(-US) dataset. More recently, Coxon et al. (2020) released the CAMELS-GB, which includes estimates of quantified discharge observation uncertainty. This dataset describes 671 catchments in Great Britain. 503 catchments (gauging stations) are complemented with quantified discharge observation uncertainty estimates (Coxon et al. (2015)). A recent effort by Kratzert et al.
(2022) combined all available national CAMELS datasets in the overarching CARAVAN dataset for global consistency and boosting accessibility through data access via Google Earth Engine.

The access to large-sample catchment data has prompted a substantial body of research, as detailed in Addor et al. (2020). This includes applications in hydrological model testing and comparative analysis (e.g., Mizukami et al. (2017); Rakovec et al. (2019); Lane et al. (2019); Feng et al. (2022)). One of the benefits of these datasets is that large samples of catchments allow
for the evaluation of the robustness of model performance (Andréassian et al. (2006); Gupta et al. (2014)). Identifying this robustness provides model users with valuable information on the presence or absence of consistency in the model results.

This study assesses the effect of omitting discharge observation uncertainty while interpreting model performance differences. Specifically, we focus on how this uncertainty influences model selection from the perspective of model users. We highlight the importance of incorporating discharge observation uncertainty during model calibration and model evaluation
efforts. To achieve this, we developed a generic method applicable to any geoscience field where model results are compared to uncertain observations. This method determines, based on the 5th to 95th percentile range of flow, if model simulation differences are significant in the context of discharge observation uncertainties. In this study, we highlight 3 use cases based on 8 hydrological models that encompass model refinement efforts, conventional model comparisons, and the influence of model structure uncertainty in light of discharge observation uncertainty. Furthermore, we assess the spatial consistency of
model performance results using a large-sample catchment dataset, and we evaluate the temporal consistency of model perfor-

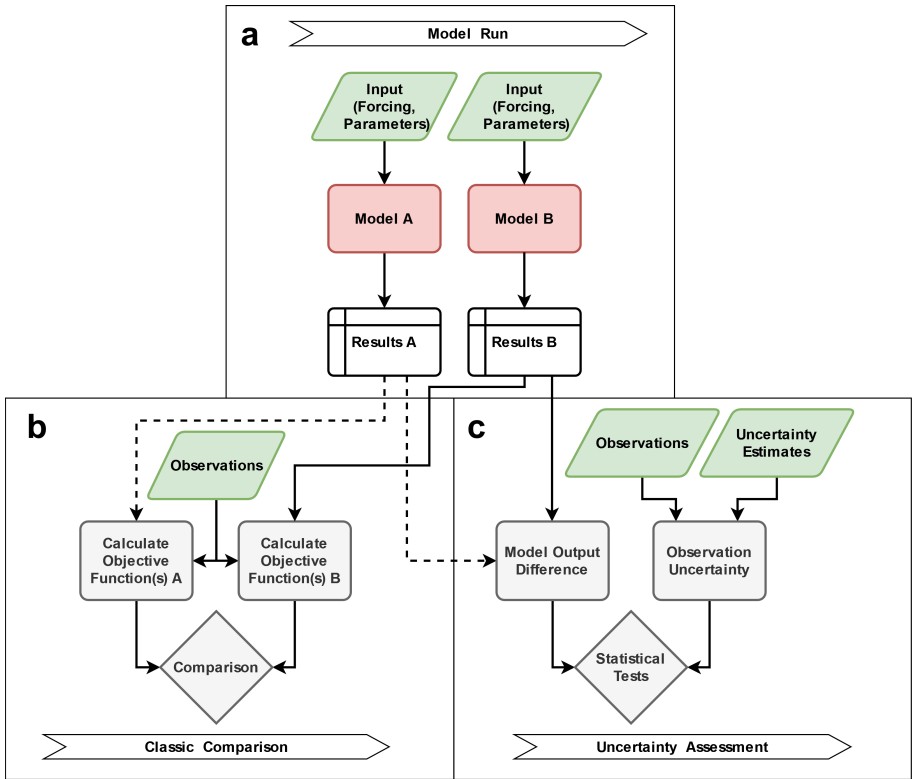

**Figure 1.** Graphical workflow of model experiments and analyses. The model experiment inputs are shown in green, the models in red, and the analysis components in grey. (a) the model runs of two models with inputs and outputs that result in simulation time series. (b) the conventional model comparison that compares objective functions based on simulation and observation time series. (c) the workflow of the proposed analysis that compares relative model simulation differences to discharge observation uncertainty estimates.

mance metrics by sub-sampling the observation and simulation pairs as demonstrated by Clark et al. (2021). By doing so, more informed conclusions can be drawn on model performance based on individual or large samples of catchments.

## 2 Methodology

A generic tooling is designed for assessing model simulations while considering the uncertainties inherent in evaluation data.
First, the 3 use cases are presented. This is followed by the input data description, evaluation data description, and the discharge observation uncertainty estimates used to conduct the analyses. Next, we describe the hydrological models and model runs employed for calibration and evaluation. The methodology concludes with an explanation of the uncertainty-based analyses.

In Figure 1, a graphical workflow provides an overview of the methodology. Figure 1a shows a typical model run with inputs and outputs, Figure 1b shows a conventional comparison of objective functions based on discharge observations and
simulations, and Figure 1c describes the uncertainty analysis introduced in this study.

## 2.1 Use cases

We devised 3 use cases based on 8 hydrological models that exemplify how users of models, who themselves are not the model developers, can interpret differences between model simulations in the context of discharge observation data uncertainty. The use cases are:

1. Model refinement in practice: This use case concerns additional model refinement by fine-tuning an effective model parameter based on discharge observation post-initial calibration. It highlights the value of relative gains in model performance when not considering discharge observation uncertainty in the calibration process.

2. Model comparison for model selection: Here, two distributed hydrological models are compared against the backdrop of uncertainties in discharge observations. This analysis aims to pinpoint scenarios where the disparities between model results are within the margin of error of the discharge observations.

3. Model selection under model structural uncertainty: This use case involves contrasting the uncertainty inherent in the model's structure, as seen across various hydrological models, with the uncertainty in discharge observations.

An additional analysis quantifies uncertainty in the model performance objective functions due to temporal sampling of the discharge simulation and observation pairs. This temporal sampling uncertainty is detailed in Section 2.5.3.

## 2.2 Data

### 2.2.1 Case study and catchment selection procedure

The CAMELS-GB large-sample catchment dataset (Coxon et al., 2020; Coxon, 2020) serves as the case study area of the use cases and contains data (hydro-meteorological time series, catchment boundaries and catchment attributes) describing 671 catchments located across Great Britain. The underlying data used to create CAMELS-GB are publicly available and, therefore, suitable for evaluating hydrological models as the dataset can be easily extended. A unique feature of the dataset is the availability of quantified discharge observation uncertainty estimates for the flow percentiles of 503 catchments (see Coxon et al. (2015)).

The use cases in this study employ hydrological models with a daily time step. This can cause temporal discretization errors in small catchments due to peak precipitation and peak discharge occurring at the same time step. Therefore, these catchments are excluded through a selection procedure. This procedure calculates the cross-correlation between observed discharge and precipitation for a range of lag times. Catchments that predominantly show less than 1 day of lag between observed discharge and precipitation are excluded. Of the 503 catchments with uncertainty estimates, 299 are selected as the case study.

### 2.2.2 Meteorological forcing and pre-processing

In this study, we use the same meteorological forcing used to create the CAMELS-GB meteorological time series and climate indices as input to the hydrological models. This input consists of gridded 1 km$^2$ daily meteorological datasets. The meteoro-

logical variables used in this study are precipitation (CEH-GEAR; Keller et al. (2015); Tanguy (2021)), reference evaporation (CHESS-PE; Robinson (2020a)), and temperature (CHESSmet; Robinson (2020b)). The distributed hydrological models use gridded inputs, and the conceptual hydrological models aggregate time series of meteorological variables readily available in the CAMELS-GB dataset.

### 2.2.3 Discharge observations and quantified uncertainty estimates

The discharge observations in the CAMELS-GB dataset were obtained from the UK National River Flow Archive and are daily values in cubic meters per second ($m^3s^{-1}$). As is common with large-sample catchment datasets, several catchments contain missing flow data in the time series. These missing values are not considered in this study's analyses.

A unique aspect of the CAMELS-GB dataset is the inclusion of quantified discharge observation uncertainty estimates created by Coxon et al. (2015). The uncertainty is quantified using a large dataset of quality-assessed rating curves and stage-discharge measurements. The mean and variance at each stage point are calculated and fitted in an iterative process using a LOWESS regression method that defines the rating curve and discharge uncertainty. Combining the LOWESS curves and variance in a Gaussian Mixture model based on a random draw from the measurement error distribution gives an estimate of discharge uncertainty; see Coxon et al. (2015) for more information.

## 2.3 Hydrological models

A mixture of distributed physically process-based and lumped conceptual hydrological models is selected for the use cases, thereby showcasing the versatility of the analysis. The model refinement and model comparison use cases employ two physically process-based hydrological models: wflow_sbm (van Verseveld et al. (2022)) and PCR-GLOBWB (Sutanudjaja et al. (2018); Hoch et al. (2023)). The rationale behind selecting these models lies in their differing approaches to conceptualizing hydrological processes and their respective optimization routines. Despite these differences, both models are suitable for comparison to a certain degree. This comparability stems from their shared classification as distributed hydrological models, similar complexity, parameterization, and applicability at a spatial resolution of 1 km$^2$.

For the model structure use case, 6 conceptual hydrological models are sourced from the Modular Assessment of Rainfall-Runoff Models Toolbox (MARRMoT: Knoben et al. (2019); Trotter et al. (2022)). These specific models are selected to encompass a wide array of model structures. The selection is based on the number of model stores, the quantity of parameters, and differing process representations.

### 2.3.1 Distributed hydrological models

The wflow_sbm physically-based distributed hydrological model (van Verseveld et al. (2022)) originated from the Topog_SBM model concept (Vertessy and Elsenbeer (1999)). This concept was developed for small-scale hydrologic simulations. The wflow_sbm model deviates from Topog_SBM by the addition of capillary rise, evapotranspiration and interception losses (Gash model; Gash (1979)), a root water uptake reduction function (Feddes and Zaradny (1978)), glacier and snow processes,

**Table 1.** Overview of the 6 selected conceptual hydrological models showing the model name, number of stores, number of parameters, and key references (adapted from Knoben et al. (2020)).

| Original Model | Number of Stores | Number of Parameters | Key References |
| --- | --- | --- | --- |
| IHACRES | 1 | 7 | Ye et al. (1997); Croke and Jakeman (2004) |
| GR4J | 2 | 4 | Perrin et al. (2003); Santos et al. (2018) |
| VIC | 3 | 10 | Liang et al. (1994) |
| XINANJIANG | 4 | 12 | Jayawardena and Zhou (2000) |
| HBV-96 | 5 | 15 | Lindström et al. (1997) |
| SMAR | 6 | 8 | Tan and O'Connor (1996) |

and D8 river routing that uses the kinematic wave approximation in this study. The parameters (40 in total) are derived from open-source datasets and use pedo-transfer functions to estimate soil properties (see hydroMT software package (Eilander and Boisgontier (2022)).

165     The 1 km$^2$ model version was aggregated from the finest available data scale (90 m). The hydraulic parameters related to the river network are upscaled using the method presented in Eilander et al. (2021). The parameter upscaling of the wflow_sbm model is based on the work by Imhoff et al. (2020) that uses point-scale (pedo)transfer-functions. This is similar to the multi-scale parameter regionalization method (Samaniego et al. (2010)). Parameters are aggregated from the original data resolution with upscaling operators determined by a constant mean and standard deviation across various scales. Fluxes and states are
170     checked for consistency during this process. See van van Verseveld et al. (2022) for further information.

The PCR-GLOBWB physically-based distributed hydrological model was initially developed for global hydrology and water resources assessments (Sutanudjaja et al. (2018)). The PCR-GLOBWB model calculates the water storage in two soil layers, one groundwater layer, and the exchange between the top layer and the atmosphere. The model accounts for water use determined by water demand. We employ the 1 km$^2$ version introduced in (Hoch et al. (2023)). The model configuration in this
175     study applies the accumulated travel time approximation for river routing.

### 2.3.2   Conceptual hydrological models

MARRMoT is a flexible modeling framework that houses an array of conceptual hydrological models Knoben et al. (2019); Trotter et al. (2022). It is particularly valued in research for assessing model structure uncertainty, as highlighted in Knoben et al. (2020). One of the key advantages of MARRMoT is that the conceptual models share a uniform numerical implementa-
180     tion. To achieve this, alterations were made to the original model codes. These alterations ensure a consistent basis for model structure comparisons, allowing for a precise evaluation of differences in hydrological simulations due to varying model structures. The hydrological models IHACRES, GR4J, VIC, XINANJIANG, HBV-96, and SMAR are selected in this study. Table 1 overviews the number of stores, parameters, and key references.

## 2.4 Model runs

The model runs that form the basis of the 3 use cases are performed as intended by the model developers. This means that this study employs calibration and or optimization methodologies as recommended by the model developers for model users. The calibrated parameters for the distributed hydrological models were obtained from the model developers. Regarding the conceptual hydrological models, we follow the model run configuration of Knoben et al. (2020).

### 2.4.1 PCR-GLOBWB model runs

The PCR-GLOBWB model does not require additional regional parameter optimization after deriving the parameter set, as this is typically not conducted by the model developers. However, this does not imply that the model would not benefit from additional optimization. The model does require a spin-up period at the start of the model run. The model is spun-up 30 years back-to-back using a single water year climatology that is based on the average values of each calendar day between 1-10-2000 and 30-09-2007. The following water year, 2008, is discarded from analyses to avoid overfitting at the start of the evaluation period, and the model is evaluated for the water years 2009 – 2015.

### 2.4.2 Default and optimized wflow_sbm model runs

The wflow_sbm model is spun-up using the water year 2000 and is additionally calibrated using discharge observations for the water years 2001-2007. Additional calibration is performed by optimizing a single parameter using the Kling-Gupta Efficiency Non-Parametric (KGE-NP) objective function (Pool et al. (2018)) based on discharge observations and simulations differences at the catchment outlet. This results in a single optimized parameter set per catchment. Imhoff et al. (2020) identified the horizontal conductivity fraction parameter (KsatHorFrac) as effective for single parameter value optimization. KsatHorFrac is an amplification factor of the vertical saturated conductivity that controls the lateral flow in the subsurface.

After calibration, the water year 2008 is discarded from analyses, and the model is evaluated for the water years 2009 - 2015. For more information on the effects of calibration, the reader is referred to Aerts et al. (2022), Section 3.1 and Figure 3. The default wflow_sbm model run sets the KsatHorFrac parameter value to the default value of 100. The calibration results of the wflow_sbm model are presented in Appendix A1.

### 2.4.3 Conceptual hydrological model runs

Similar to the other model runs, the conceptual hydrological model runs are spun-up using the water year 2000 and calibrated using the water years 2001-2007. The calibration method uses the Covariance Matrix Adaptation Evolution Strategy (CMA-ES; Hansen et al. (2003); Hansen (2006); Hansen and Ostermeier (2001)). This method optimizes a single-objective function to find global parameter optimums based on non-separable data problems. A demonstration of the sensitivity of the calibration parameters is shown in Knoben et al. (2020). Following calibration based on the KGE-NP objective function, the water year 2008 is discarded. The models are evaluated based on the water years 2009-2015 using the same KGE-NP objective function.

#### 2.4.4 eWaterCycle

This study is conducted using the eWaterCycle platform (Hut et al., 2022). eWaterCycle is a community-driven platform for running hydrological model experiments. All components that are required to run hydrological models are FAIR by design (Wilkinson et al., 2018). This is achieved by versioning models and datasets and creating reproducible workflows. Therefore, the platform is suitable for conducting model performance experiments. The model simulations were conducted on the Dutch supercomputer Snellius to ensure faster model run time. We created example notebooks that use the eWaterCycle platform on cloud computing infrastructure: https://doi.org/10.5281/zenodo.7956488.

### 2.5 Analyses

#### 2.5.1 Model evaluation

The hydrological model runs (calibration and evaluation) are evaluated using the Kling-Gupta efficiency non-parametric (KGE-NP, Pool et al. (2018)) objective function. This efficiency metric deviates from the more commonly used Kling-Gupta efficiency (KGE, Gupta et al. (2009)) by calculating the Spearman rank correlation and the normalized-flow-duration curve instead of the Pearson correlation and variability bias. Values range from $-\infty$ to 1 (perfect score). In addition to the KGE-NP metric, we consider the Nash-Sutcliffe efficiency (NSE, Nash and Sutcliffe (1970)) to demonstrate the sensitivity of the results towards the selection of objective function. We include the KGE-NP, KGE, modified KGE (Kling et al. (2012)), and the NSE objective functions in the data repository for completeness and future reference.

#### 2.5.2 Discharge observation uncertainty

The ad hoc discharge observation uncertainty-based analysis of model performance differences consists of 3 parts. The first part divides the observation and simulation pairs into 3 flow categories similar to Coxon et al. (2015): low flow, average flow, and high flow conditions. The low flow category is based on the observed discharge values at the catchment outlet between the 5th and 25th percentile range, average flow on the 25th to 75th percentile range, and high flow on the 75th to 95th percentile range. Not all percentiles are included for the low and high flow categories due to limited data availability on quantified discharge observation uncertainty.

In the second part of the method, shown in Figure 1c, the first step is calculating the absolute error between calibrated model simulations for each flow category and each catchment. This step is visualized as a hydrograph in Figure 2a (blue line) and is referred to as the model difference. Next, the discharge observations' upper and lower uncertainty bounds are taken from the CAMELS-GB dataset. For example, the upper uncertainty percentages of a flow category boundary percentiles correspond to 25 % and 15 %, respectively. These values are then averaged to obtain the upper uncertainty bound at 20 % discharge observation uncertainty, as shown by the orange line in Figure2b. Similarly, the red line in Figure2b is the calculated lower uncertainty bound. The observation uncertainty percentage is then calculated by taking the absolute average of both uncertainty

bounds (17.5 %). This percentage is applied to the observations to quantify the portion of the observed discharge attributed to uncertainty (observation uncertainty) in cubic meters per second ($m^3s^{-1}$), as illustrated by the green line in Figures 2bc.

The third part of the method applies a dependent t-test using the time series in Figure 2c with a 0.05 significance level to determine if the observation uncertainty time series is greater than the model simulation difference time series.

This method is subject to certain limitations, particularly regarding using the discharge observation uncertainty estimates. Due to the absence of data, the upper and lower 5th percentiles of flow could not be included, while these data points can be most important for users to determine the fit-for-purpose of a model. In addition, using the rating curve uncertainty rather than the uncertainty bounds of flow percentiles is preferred. We accept these limitations as we promote the use of existing datasets to ensure community participation in implementing the suggested evaluation procedure in other studies.

### 2.5.3   Temporal sampling uncertainty

Another aspect of model performance evaluations that might misinform model users is the sensitivity of objective functions to the temporal sampling of time series. Temporal sampling uncertainty determines if the error distribution of simulation and observation pairs is heavily skewed. A few data pairs might have a disproportionate effect on the calculated objective functions that are used to determine model performance. The inclusion or exclusion of these data points due to the selection of the calibration and evaluation period alters the consistency of model performance over time.

To quantify the temporal sampling uncertainty of the KGE-NP objective function, we applied the methodology of Clark et al. (2021). This method sub-samples the simulation and observation time series through bootstrapping and (Efron, 1979) and jackknife-after-bootstrap (Efron and Tibshirani, 1986) methods. The change in objective function due to the shuffling of the sub-samples allows for calculating the standard error and its tolerance interval. The tolerance intervals corresponding to each model instance are averaged and referred to as the temporal sampling uncertainty. We extended the GUMBOOT software package Clark et al. (2021) by adding the KGE-NP metric for this study.

## 3   Results

In this section, we first present an overview of the discharge-based model performance results for each of the 3 use cases. Next, we detail the spatial distributions of the maximum model performance difference. This is followed by presenting uncertainty estimates for discharge observations categorized by flow. Next, the discharge observation uncertainty-based relative model performance analyses are presented. The section ends with the temporal sampling uncertainty analysis results.

Appendix A.1 contains the calibration results of the wflow_sbm model, and Appendix A.2 the Nash-Sutcliffe efficiency (NSE) based model performance results of all considered models.

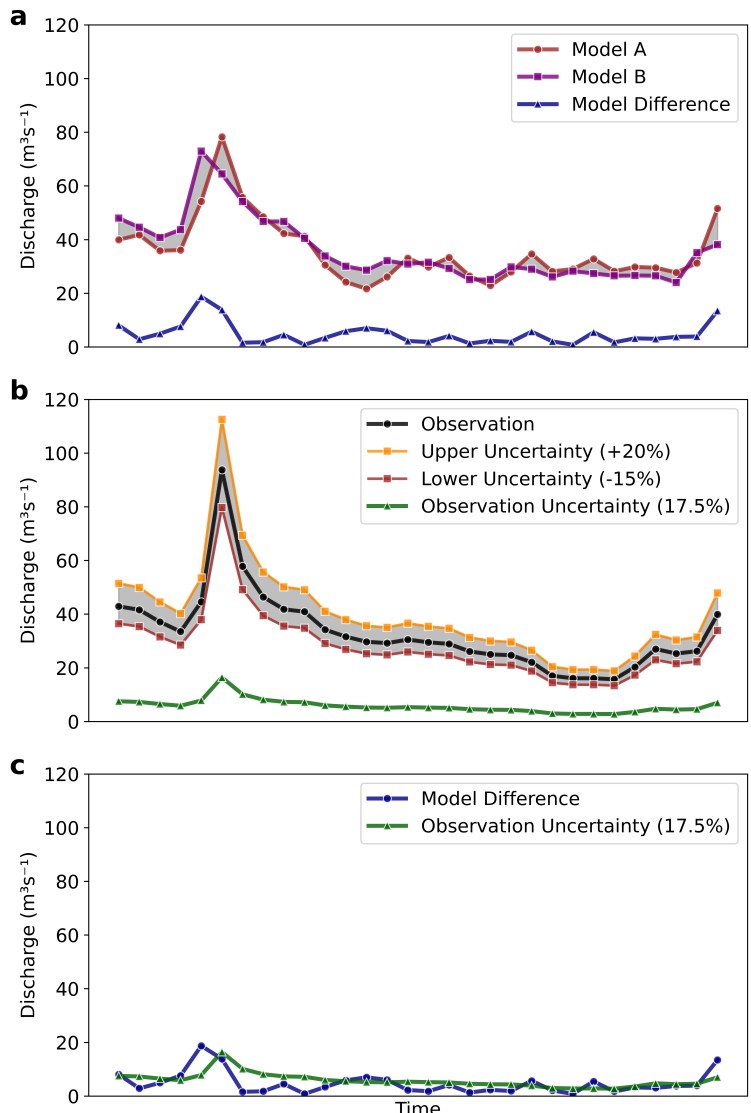

**Figure 2.** Example hydrographs of the discharge observation uncertainty analysis method. (a) calculation of the absolute difference (blue) between calibrated model simulations (red and purple). (b) calculation of streamflow observation uncertainty in $m^3s^{-1}$, shown in green. Orange and dark red lines indicate upper and lower bounds of observation uncertainty percentages that are averaged and multiplied with the observations in black. (c) resulting time series, with, in blue, the absolute difference between calibrated model simulations and, in green, the averaged discharge observation uncertainty in $m^3s^{-1}$.

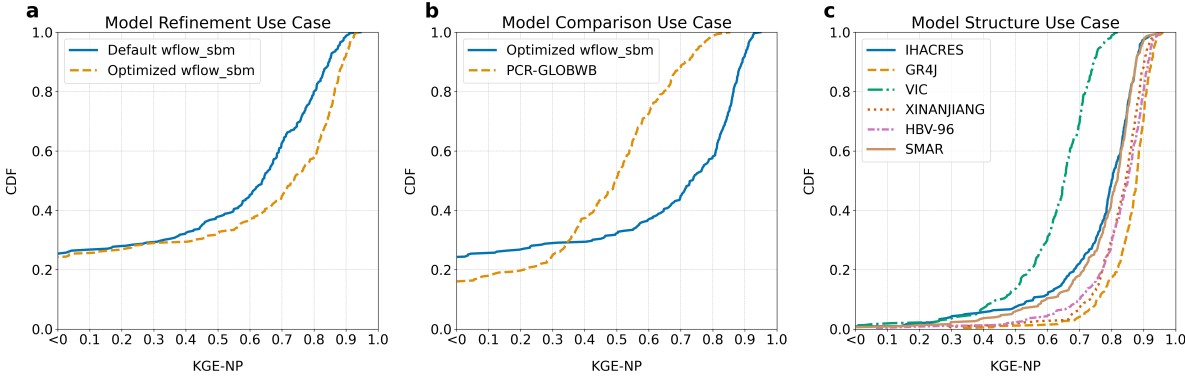

**Figure 3.** Cumulative distribution function (CDF) plots of the Kling-Gupta Efficiency non-parametric (KGE-NP) objective function, derived from discharge estimates and observations at 299 catchment outlets. (a) shows the CDF for the model refinement use case, optimizing the wflow_sbm hydrological model with a single parameter. (b) shows the CDF for the model comparison use case, comparing the optimized wflow_sbm and PCR-GLOBWB hydrological models. (c) demonstrates the CDF for the model structure use case, showcasing results from 6 conceptual hydrological models.

### 3.1 Discharge-based model performance

Model performance is assessed using discharge observation and simulations at 299 catchment outlets. The results are shown in Figure 3 as Cumulative Distribution Functions (CDFs) of the KGE-NP objective function. These results offer insight into the
model's accuracy in simulating observed discharge.

The CDF of the model refinement use case in Figure 3a establishes that optimizing a single effective parameter improves approximately 65% of the catchment simulations. The improvements remain modest, as indicated by the median value of 0.64 KGE-NP for the default wflow_sbm model and 0.74 KGE-NP for the optimized wflow_sbm model. Larger model performance differences are found for the model comparison use case in Figure 3b. Here, the optimized wflow_sbm model performs better
in 75% of the catchments than the PCR-GLOBWB model. Both models demonstrate poor results for between approximately 18 % and 24% of the evaluated catchments (<0.40 KGE-NP).

The model structure use case results are based on 6 conceptual hydrological models that only deviate in model structure (Figure 3c). The spread in model results shows that the VIC model lags behind in performance compared to the other models. The IHACRES and SMAR models yield very similar results despite large structural differences. The XINANJIANG and HBV-
96 models produce comparable outcomes and share a more similar model structure. The GR4J model consistently outperforms the other models. The total model structure uncertainty, as expressed by the difference between the worst and best-performing model's CDF, is substantial, while the differences between models can be subtle. Median KGE-NP values for the models are as follows: VIC at 0.65, IHACRES at 0.80, SMAR at 0.82, XINANJIANG at 0.84, HBV-96 at 0.85, and GR4J at 0.88 KGE-NP.

Next, we consider the spatial distribution of the results, as shown in Figure 4, which depicts the maximum KGE-NP differ-
ence between the models for each use case. Figure 4a indicates improvements after model refinement, with positive KGE-NP

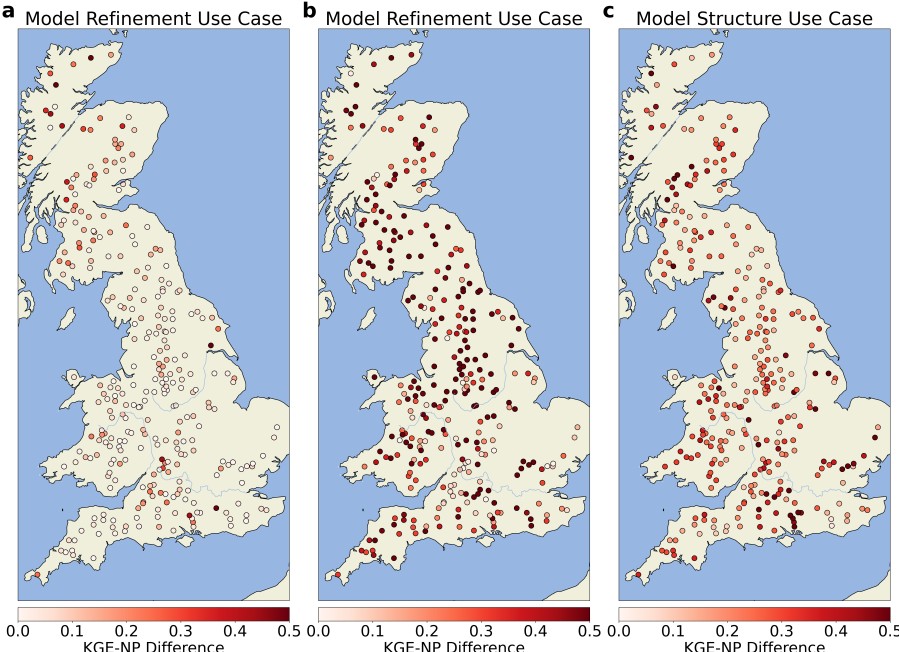

**Figure 4.** Spatial distribution of the absolute Kling-Gupta Efficiency non-parametric (KGE-NP) objective function difference between the worst and best model's performance per catchment and use case. (a) the model refinement use case is based on the default and optimized wflow_sbm hydrological models. (b) the model comparison results are based on the optimized wflow_sbm and PCR-GLOBWB hydrological models. (c) the model structure use case results are based on the worst and best model performances of the 6 conceptual hydrological models.

difference values in various parts of Great Britain. However, there are no discernible spatial trends in these improvements. Figure 4b compares the wflow_sbm and PCR-GLOBWB distributed hydrological models, revealing high spatial variability with no consistent patterns. Similarly, Figure 4c highlights differences for the model structure use case, where the largest differences are again observed in various regions without a clear spatial trend. While the spatial distribution is provided for completeness, 295 no significant spatial trends are evident from the data.

## 3.2 Discharge observation uncertainty estimates

The discharge observation uncertainty estimates consider the 5th to 95th percentile range of flow. These estimates are categorized into 3 flow conditions and are presented in Figure 5. In the box plot for the low flow category, we observe a wide interquartile range, shown by the spread of the box. This indicates a higher variability in discharge observation uncertainty 300 percentages. The median value, represented by the line within the box, is at the 20% uncertainty mark. Many outliers above the box indicate occasional large deviations from the median value. The range of values for the average flow category is narrower than for the low flow category, with a median value of 15%. The lowest median value is found for the high flow category at

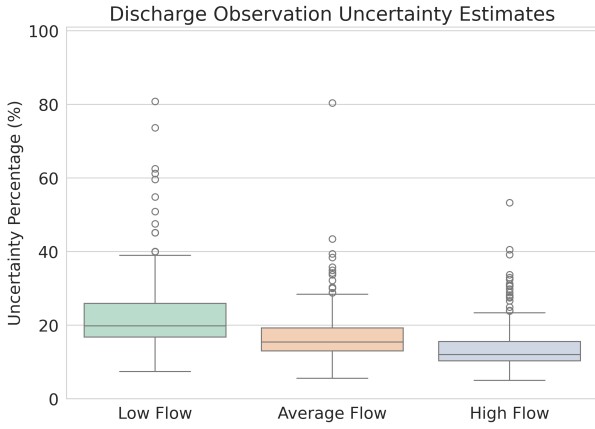

**Figure 5.** Discharge observation uncertainty estimates of 299 catchment outlets based on the work of Coxon et al. (2015) expressed as uncertainty percentages per flow category. (a) the low flow category uncertainty estimates are based on the 5th to 25th flow percentiles. (b) presents the 25th to 75th percentile average flow category. (c) the high flow discharge observation uncertainty estimates of the 75th to 95th flow percentiles are shown.

12%. It is important to mention that the uncertainty is expected to be considerably higher if the underlying data contains the upper 5th percentiles of flow for this category.

### 3.3 Use Cases

The discharge simulation difference time series of two models is expressed in cubic meters per second and compared to the discharge observation uncertainty time series in cubic meters per second. This is done using a t-test to determine if the simulation differences are larger than the discharge observation uncertainty estimates. The instances where this is the case are reported in Table 2 for the 3 use cases.

#### 3.3.1 Model refinement

The model refinement use case results in Table 2 show that approximately one-third of the considered catchments contain instances of simulation differences between the wflow_sbm default and wflow_sbm optimized models that are statistically smaller than the discharge observation uncertainty estimates. This demonstrates the importance of incorporating (discharge) observation uncertainty when performing model refinement, especially based on a large-sample catchment dataset. This con-

sideration should be part of the calibration and subsequent evaluation process. In addition, the results indicate that it is difficult to conclude whether the model performs better after refinement when discharge observation uncertainty is not considered. Overall, the results affirm the importance of incorporating discharge observation uncertainty in the optimization routine of the wflow_sbm model.

**Table 2.** Overview of the number of instances per flow category where discharge observation uncertainty exceeds the simulation differences based on 299 catchments. Results are based on dependent t-tests with a significance level of 0.05.

| Use Case | Models | Flow Category | Discharge Obs. Uncertainty > Model Sim. Difference |
|---|---|---|---|
| Model Refinement | wflow_sbm Default & Optimized | Low | 98 |
| | | Average | 98 |
| | | High | 115 |
| Model Comparison | wflow_sbm Optimized & PCR-GLOBWB | Low | 5 |
| | | Average | 4 |
| | | High | 3 |
| Model Structure | 6 Conceptual Hydrological Models | Low | 1 |
| | | Average | 0 |
| | | High | 0 |

### 3.3.2 Model comparison

For the model comparison use case (Table 2), there is a lower frequency of instances where discharge observation uncertainty surpasses differences in discharge simulations. The comparison between the optimized wflow_sbm model and the PCR-GLOBWB model reveals that simulation differences exceed discharge uncertainty estimates in 5 catchments for low flow, 4 for average flow, and 3 for high flow categories. These findings suggest that the interpretation of model performance is not significantly affected by the ad-hoc addition of discharge observation uncertainty. However, catchments demonstrating the impact

of observation uncertainty warrant careful examination.

### 3.3.3 Model structure

The analysis of model structure uncertainty in the context of discharge observation uncertainty reveals that only a single instance of the low flow category contains discharge observation uncertainty that exceeds the simulation difference between all 6 conceptual hydrological models (Table 2). This establishes that based on the selected models, the model structure uncertainty,

expressed as the difference in discharge simulations, is larger than this dataset's discharge observation uncertainty estimates. However, the investigation into the differences between the individual models yields several insights based on the results in Figure 6.

The VIC model results, characterized by its relatively lower performance, contain only a few instances where discharge observation uncertainty exceeds simulation differences, making it identifiable as the lesser-performing model. In contrast, the

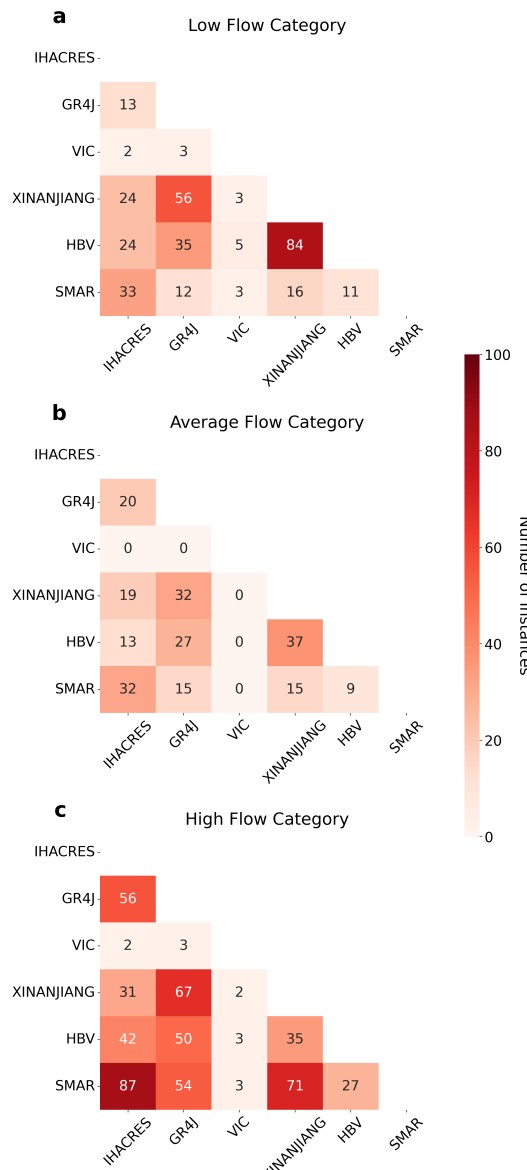

**Figure 6.** Heat map of the 6 conceptual hydrological models showing for each model combination the number of instances (n=299) that discharge observation uncertainty exceeds simulation differences. (a) number of instances for the low flow category, with low values shown in white and high values in red. (b) number of instances for the average flow category. (c) number of instances for the high flow category.

IHACRES and SMAR models exhibit a high level of simulation agreement, demonstrated by a large number of instances in Figure 6c. This is despite significant differences in their complexity and structural design. Namely, IHACRES is a single-store hydrological model, and SMAR is a 6 store hydrological model that accounts for soil moisture in a separate store.

This alignment of simulation results between models with varying complexities highlights the nuanced influence of structural differences on simulation outcomes. The HBV-96 and XINANJIANG models most resemble each other regarding the number of stores and parameters; the parameters themselves and process descriptions differ. Both models contain a low number of instances, allowing the identification of the better-performing model.

Next, we examine the results across the individual flow categories. The low flow category (Figure 6a) and the average flow category (Figure 6b) show similar trends, though with a lower number of instances for the average flow category with a lower number of instances for the average flow category. The high flow category (Figure 6c) is characterized by a more frequent discharge observation uncertainty surpassing simulation differences. This is especially evident between the IHACRES and SMAR models. The variability in structural design and parameterization among different hydrological models leads to notable output differences. This underscores the importance of selecting the appropriate model by including discharge observation uncertainty in the calibration and evaluation process.

### 3.4 Temporal sampling uncertainty

The temporal sampling uncertainty of the KGE-NP objective function is defined as the tolerance interval of the standard error of the objective function due to the sub-sampling of the simulation and observation pairs. This analysis provides insights into hydrological model performances' temporal reliability and interpretability. Analysis of results from the 6 conceptual hydrological models, as shown in Figure 7b, reveals a pattern consistent with the model performance depicted in Figure 3c. Precisely, the VIC model displays the highest KGE-NP uncertainty across all catchments, indicating its variability and the challenges in using this model's current setup for accurate predictions in different hydrological contexts.

The IHACRES and SMAR models and the GR4J, XINANJIANG, and HBV-96 model groups show similar KGE-NP-based temporal sampling uncertainty levels. This consistency across models with varying complexities suggests that KGE-NP uncertainty is influenced not only by the model design but also by hydrological conditions and data quality. Uncertainty values range widely, from about 0.1 KGE-NP to over 0.6, indicating significant variability in the temporal robustness of results (Figure 7b).

When comparing the average KGE-NP objective function uncertainty with the KGE-NP differences between individual models, it becomes clear that uncertainty often overshadows the differences between models. This is particularly the case in comparisons between GR4J - HBV-96, XINANJIANG - HBV-96, and SMAR - IHACRES. These findings imply that the inherent uncertainty in the objective functions may limit the ability to distinguish between model performances, complicating efforts to identify the most fit-for-purpose model based on this metric alone. This underscores the need for a more nuanced approach to model evaluation that considers objective function metrics, other contextual factors, and additional performance measures, ensuring more robust and reliable model selection processes.

### 4 Discussion

We introduced an ad hoc method highlighting the importance of including discharge observation uncertainty when evaluating hydrological models. Discharge observation uncertainty is frequently overlooked by model users, leading to potential misinter-

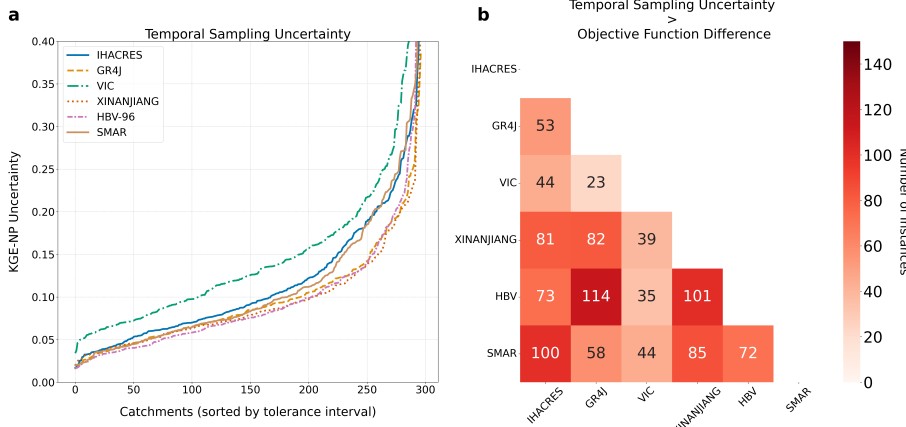

**Figure 7.** (a) Temporal objective function sampling uncertainty based on 6 conceptual hydrological models expressed as the average tolerance interval of the standard error due to sub-sampling. On the horizontal axis, the sorted values per catchment, and on the vertical axis, the KGE-NP objective function uncertainty. (b) Heat map of the 6 conceptual hydrological models showing for each model combination the number of instances (n=299) that the average objective function uncertainty exceeds the objective functions differences of model combinations. Low values in white and high values in red indicate the number of instances.

pretations of relative model performance. Our findings emphasize the significant impact of discharge observation uncertainty on model performance interpretation.

We acknowledge that observation uncertainty is not the only source of uncertainty as there are uncertainties in model inputs, model structure, parameter sets, and initial or boundary conditions (e.g., Renard et al. (2010); Dobler et al. (2012); Hattermann et al. (2018); Moges et al. (2021)). Therefore the proposed generic tooling does not replace a complete uncertainty analysis of modeling chains that also accounts for the impact of input uncertainties (Beven and Freer (2001); Pappenberger and Beven (2006); Beven (2006)). Instead, it assists model users in interpreting relative model performance and highlights the importance of conducting a complete uncertainty analysis. Therefore, our study only constitutes a fraction of a broader challenge in which input uncertainty plays a substantial role, as demonstrated in Bárdossy and Anwar (2023).

### 4.1 Performance interpretation under discharge observation uncertainty

Our analysis demonstrates that regionally optimizing the wflow_sbm hydrological model often results in only marginal improvements in model performance (Figure 3a). Although any improvement is beneficial, the findings suggest that discerning the superior model variant becomes challenging without factoring in the uncertainty of discharge observations during calibration. This is evident in 98 instances of low and average categories of flow and 118 cases of the high flow category (Table 2). The number of instances is expected to increase further when including flows of the lower and upper 5th percentiles of flow are included. Adopting an ad hoc measure, as introduced in this study, provides a practical but limited method for improving the

interpretability of relative model results. Therefore, we recommend the integration of discharge observation uncertainty into both the model calibration and evaluation procedures, aligning with the consensus in the literature.

When comparing different hydrological models, we find that the uncertainty of discharge observations slightly masks the differences in relative model performance as shown by the 3 to 5 instances per flow category in Figure 3b and Table 2. Similarly
to the model comparison use case, the model structure use case indicates that structural uncertainties overshadow the effects of discharge observation uncertainty. However, the comparison of individual models in Figure 6 shows many instances of discharge observation uncertainty exceeding model performance differences. For example, despite their structural differences, the IHACRES and SMAR models demonstrate a high level of simulation agreement (Figure 3c) and subsequent difficulty in discerning model performance differences in light of discharge observation uncertainty. In contrast, the VIC and XINANJIANG
models, which have similar structures, exhibit two-thirds of the catchment simulation differences within the uncertainty bounds of the discharge observations. This underlines the complex interplay between model structures and subsequent performance, especially when contrasted with the uncertainty of discharge observation.

## 4.2    Temporal robustness of model performance

Model performance can be heavily influenced by a few data points in the time series on which model performance is based
(Clark et al. (2021)). This can result in biased model performance interpretations depending on the selected time period for calibration and evaluation. When models are sensitive to specific data points, this can be due to inadequate process descriptions for the considered models. In addition, this might also indicate the presence of disinformative events and model invalidation sites where the runoff coefficient exceeds a value of 1 (Beven and Smith (2015); Beven (2023); Beven and Lane (2022); Beven et al. (2022)) or the presence of atypical data (e.g. Thébault et al. (2023)).
Models ought to demonstrate adequate performance across the entire time series, which should be accurately represented in the performance outcomes. The assessment of temporal sampling uncertainty does not imply that this should not be the case; it instead points towards the model simulation and observation pairs that are worth investigating. These instances can serve as indicators that suggest areas where models may require further scrutiny and improvements. Knowing the temporal sampling uncertainty is relevant for model users as it provides information on the consistency of the model performance over time, which
is necessary to determine the fit-for-purpose of a model. Therefore, it is recommended to include alternative estimators better suited for skewed performance data in the reporting of model performance (e.g., Lamontagne et al. (2020); Shabestanipour et al. (2023); Towler et al. (2023)).

## 4.3    Practical implications for model users

The method introduced in this study is purposely designed to be as generic and straightforward as possible to increase the
potential for adoption in future studies. It can be applied to any hydrological state or flux where observation time series include uncertainty estimates. In addition, we recommend the routine reporting of evaluation data uncertainties and the temporal sampling uncertainty of objective functions. This would yield a clearer understanding of the relevance of differences between model outcomes and aid in identifying samples that require cautious interpretation. This reporting, however, does not replace

model benchmarks that include complete uncertainty analyses (e.g., Lane et al. (2019)) but enhances the interpretability of model performance in its absence.

For model users, this approach offers a pragmatic way to understand the implications of uncertainty in their model selection processes. While our method facilitates a clearer understanding of where and how uncertainties affect relative model performance differences, it should be viewed as a complementary step rather than a replacement for a thorough uncertainty analysis.

## 4.4 Limitations & Outlook

The study presented faces several practical limitations. First, excluding the lower and upper 5th percentiles of flow from the analysis introduces a constraint on the uncertainty assessment, overlooking critical flow conditions often of significant interest in hydrological studies. This exclusion limits the ability to fully understand model performance under a complete range of hydrological conditions. Second, the reliance on uncertainty bounds rather than direct uncertainty estimates from rating curves, due to their absence in the CAMELS-GB dataset, poses another limitation. Using broad uncertainty bounds instead of precise estimates derived from rating curves, the analysis may not capture the variability and uncertainty inherent in the discharge observations. Additionally, displaying discharge observation uncertainty as relative values in the form of uncertainty percentages in Figure 5 might imply that the categories are affected at the same rate by the same physical phenomena. For instance, the largest changes to the rating curve for high flows are caused by changes in the river width, while low flow is most sensitive to sedimentation. Therefore, small changes in flow volume might have a larger effect on low flow conditions than high flow conditions. Expressing the values as absolute values, as used in the methodology of this study, is not feasible due to large differences in flow volumes between catchments, a common problem for large-sample catchment studies. Last, the study focuses solely on evaluating model performance primarily through discharge simulations, without delving into the reasons behind good or poor model performance, as this is outside the study's scope.

Looking forward, addressing discharge uncertainty on a global scale is of paramount importance. Accurate global assessments of discharge uncertainty are critical for informing water management strategies, policy decisions, and climate impact studies. Understanding and mitigating these uncertainties can develop more reliable hydrological models and enhance resource management worldwide. Although this study only provides a glimpse of what this might imply, it highlights the necessity for such global assessments by incorporating discharge observation and temporal sampling uncertainties into hydrological evaluations.

## 5 Conclusions

This study assesses the importance of including discharge observation uncertainty and temporal sampling uncertainty of objective functions in hydrological model performance evaluations based on a large-sample catchment dataset. This is done by statistical testing that determines if the difference in discharge simulations between two hydrological models is larger or smaller

than the discharge observation uncertainty estimates. Flow categories are created between the 5th and 95th percentile range of observed flow to support this analysis, and 3 use cases are devised.

In the model refinement use case, 100 out of 299 catchment instances showed discharge simulation differences between the default and optimized wflow_sbm models within the uncertainty bounds of discharge observations. This emphasizes integrating discharge observation uncertainty into the calibration process for model refinement. As a result, it is difficult to discern if

the optimization of the model leads to improved simulations of actual discharge. For the model comparison use case, we found that depending on the model combinations, a large fraction of catchments showed discharge observation uncertainty exceeding simulation differences. This suggests careful consideration of this uncertainty in model performance evaluations. The model structure uncertainty use case based on 6 conceptual hydrological models indicated only a few instances of discharge observation uncertainty exceeding simulation differences. Indicating that model structure uncertainty, expressed as discharge sim-

ulation differences, often exceeds discharge observation uncertainty. A comparison of the six individual hydrological models showed no clear relation between model complexity and model performance. Our study underscores the necessity of integrating discharge observation uncertainty and temporal sampling uncertainty into hydrological model evaluations to ensure accurate, reliable, and meaningful assessments of model performance. Implementing our proposed methodology in reporting practices is expected to improve the robustness of hydrological model result interpretation, aiding in more informed model selection and

refinement decisions by model users.

Furthermore, addressing uncertainty on a global scale is of paramount importance. Accurate global assessments of discharge uncertainty are critical for informing water management strategies, policy decisions, and climate impact studies. Understanding and mitigating these uncertainties can lead to more reliable hydrological models and better resource management worldwide. This study provides a foundation for such global assessments by demonstrating the necessity of incorporating both discharge

observation and temporal sampling uncertainties into hydrological evaluations.

*Code availability.* https://github.com/jeromaerts/CAMELS-GB_Comparison_Uncertainty, https://doi.org/10.5281/zenodo.7956488

*Author contributions.* JPMA wrote the publication. JPMA, JMH, and RWH, conceptualized the study. JPMA, JMH, RWH, NCvdG, GC developed the methodology. JPMA, JMH, conducted the analyses. JMH, RWH, NCvdG, GC did internal reviews. RWH, NCvdG are PIs of the eWaterCycle project.

*Competing interests.* The contact author has declared that none of the authors has any competing interests.

*Acknowledgements.* This work has received funding from the Netherlands eScience Center (NLeSC) under file number 027.017.F0. We would like to thank the research software engineers (RSEs) at NLeSC who co-built the eWaterCycle platform and Surf for providing computing infrastructure. Gemma Coxon was supported by a UKRI Future Leaders Fellowship award [MR/V022857/1].

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

# 1 A.1 wflow_sbm calibration

# A1 A.2 NSE based model performance results

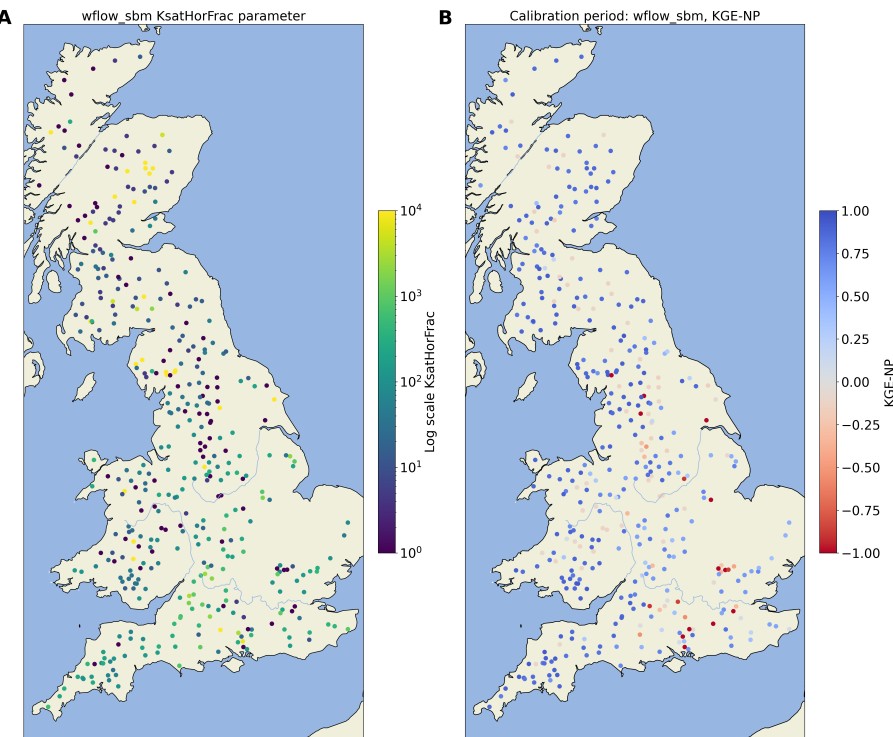

**Figure A1.** (A) Spatial distribution of the best performing KsatHorFrac calibration parameter of the wflow_sbm model based on additional calibration on discharge observations. (B) Spatial distribution of the KGE-NP objective function based on the calibration period of the wflow_sbm model.

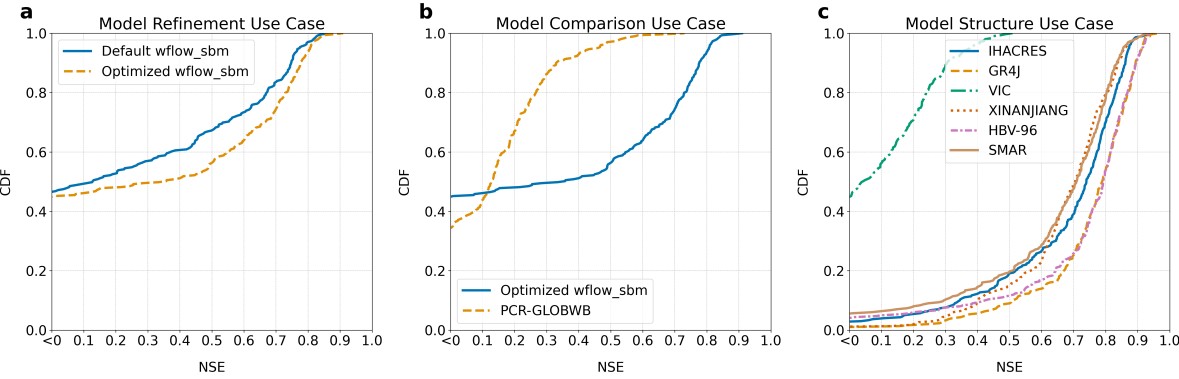

**Figure B1.** Cumulative distribution function (CDF) plots of the Nash-Sutcliffe Efficiency (NSE) objective function, derived from discharge estimates and observations at 299 catchment outlets. (a) shows the CDF for the model refinement use case, optimizing the wflow_sbm hydrological model with a single parameter. (b) shows the CDF for the model comparison use case, comparing the optimized wflow_sbm and PCR-GLOBWB hydrological models. (c) demonstrates the CDF for the model structure use case, showcasing results from six conceptual hydrological models.