# Peer review of "On the importance of discharge observation uncertainty when interpreting hydrological model performance"

_EGUsphere, 2023_

## Author Comment (AC1)

Reply to Review #1

First and foremost, we would like to express our sincere gratitude to Prof. Beven for the valuable insights and constructive comments on our publication.

We feel that the review may not fully capture the key message and central focus of our work, indicating that our framing requires significant improvement. In particular, we need to clarify the distinction between model developers and model users, a vital aspect for the proper framing of this study. As the authors of this publication, we belong to the model user category.

It is essential to clarify that our intention is not to overlook any structural issues in the inputs, models, or their applications. The central point of our study is: "How do users of models, whom are not themselves the developers of these models" have to interpret differences in model output in the light of discharge observation uncertainty. Our study provides these users of models with the workflow and statistical tools to make that decision. We make a clear distinction between model developers and model users, because in daily use of models this distinction is also present. We illustrate this workflow and its usefulness with the model runs done with the two models on the CAMELS-GB dataset. The models are employed in this study as intended by the developers to demonstrate inevitable short comings and strengths of these models. We recognize the need to address this framing issue throughout the publication after careful revision.

In the following section, we will provide a point-by-point response to each concern and outline the steps we will take to address them adequately.

Point-by-point response:

*"Observation uncertainties should not be separated from model calibration as has been done here – since any optimal parameter set depends on the particular sequence of errors in the observations (as well as whatever objective function is used). In fact it is pretty obvious that allowing for ANY source of observation error as part of the calibration process means that the concept of an optimal parameter set has no real meaning (as discussed at least since 1989)."*

We agree with the importance of incorporating observation uncertainties in model calibration, as emphasized by Beven & Binley (1992), Beven & Freer (2001), Beven & Smith (2015), McMillan et al., 2018, Beven & Lane (2019), and Westerberg et al. (2022). It is crucial to note that the hydrological models employed in our study were initially calibrated by the model developers. As users, our primary focus is not solely on highlighting shortcomings in the models. Instead, our main emphasis lies in identifying and highlighting instances where differences between model outputs fall within the bounds of streamflow observation uncertainties.

We appreciate the reviewer's attention to this aspect and will ensure that the framing and messaging of our publication clearly reflect the objectives of our study. In addition, we will include the necessary literature to support this view in the discussion section of the publication.

*"Input uncertainties should not be ignored, as has been done here, since they can be an important source of disinformation in calibration data sets."*

We appreciate the reviewer's acknowledgment of the importance of considering input uncertainties in the calibration process. The works of among others Kavetzki et al. (2006a, 2006b) and Westerberg et al. (2022) indeed highlight the significance of not disregarding input uncertainties. It is true that including input uncertainties in the calibration process is important when aiming to demonstrate the best possible hydrological model setup. Given the focus of this publication and the absence of input uncertainties (i.e. rainfall and PET) for large-sample hydrology studies, we employed the perspective

that using the same inputs for both models can help minimize the impact of these uncertainties. By forcing both models with the same set of uncertainties, we create a more direct comparison. Therefore, we promote the use the same forcing inputs for other hydrological modelling efforts that use the CAMELS-GB dataset as the case study.

We want to emphasize that the consideration of input uncertainties remains an important aspect that should not be overlooked. While using the same inputs for both models helps reduce the potential bias introduced by different input assumptions, it is still valuable to acknowledge and discuss the implications of these uncertainties in the context of model comparison and evaluation. In the revised version of our publication, we will make sure to explicitly address the role of input uncertainties in the discussion. In addition, we will use the term "observed discharge uncertainty" instead of " observation uncertainty" throughout the revised manuscript.

*"The authors refer to "outliers", but the authors do not differentiate between whether outliers might be the most important events in distinguishing models (e.g. Singh and Bardossy, AWR 2012), or whether they might in some cases introduce disinformation into the calibration process (e.g. the runoff coefficients greater than 1 of Beven and Smith, JHE ASCE 2015; Beven, PRSL 2019; Beven et al., HP 2022, 2023)."*

We acknowledge the concerns raised regarding the term 'outliers' and its potential negative connotation in the context of science and hydrology. Therefore, we will consider adopting an alternative term to refer to these data points. Denoting 'outliers' with 'heavy tails' instead, better describes the presence of heavy tails in the residuals of the squared error distribution of the observation and simulation pairs. In addition, we want to clarify that our intention is not to downplay the importance of these data points. We rather emphasize the need for careful inspection before drawing conclusions based solely on streamflow-based model performance. In our publication, these data points represent the heavy tails of the residuals between observation and simulation pairs. It is important to note that these points can have a disproportionally significant effect on the objective function, as discussed in Clark et al. (2021). As highlighted in the review, these data points can indeed often hold important information and serve as the most important events in distinguishing models and the introduction of disinformation.

To address this issue more comprehensively, we will adjust the terminology to 'heavy tails' in the introduction and discussion sections of our paper. By more critically reflecting on the nature and implications of these specific data points, we aim to present a clearer and more accurate representation of our findings. Additionally, we will further support our discussions with references to relevant literature such as Lamontagne et al. (2020) and Shabestanipour et al. (2023).

*"Is the temporal sampling issue (in terms of different sampling periods) really relevant? Should we not give models the maximum chance to fail over extremes (after assessing for possible disinformation) by using as much data as possible (Shen et al., WRR 2022 also recently suggested this as the most robust calibration for use in prediction)."*

We agree that giving models the maximum chance to fail over extremes is fair.  We achieve this by evaluating the models over a prolonged period. Nevertheless, we must not overlook the relevance of temporal sampling when comparing hydrological models. As pointed out by Fowler et al. (2018), equally weighting each year in the calibration data emerges as the most robust strategy. This approach ensures that a given wet year's influence, that may have a significant impact on the KGE-NP score, is balanced with the potential value contained in dry years, particularly in the context of a changing climate. Furthermore, the work of Lamontagne et al. (2020) highlights an essential aspect of certain objective functions, which might exhibit low bias but still manifest considerable variability between samples of the streamflow time series. This variability can stem from the skewness and periodicity of the streamflow data.

Therefore, we argue that it is indeed important to demonstrate and account for the impact of temporal sampling when performing extensive hydrological model evaluations. By considering this aspect, we can obtain a more comprehensive and reliable understanding of model performance across diverse hydrological conditions.

*"There are, however, other temporal sampling issues in terms of the discretisation error of using daily time steps on some rather small UK basins. This really should be taken into account in that for some events it might be more significant than the rating curve error depending if an observed peak falls on one day or another relative to the model prediction."*

Indeed this is clearly present in the hydrological model results. We will more critically reflect on this in the results and discussion of the publication. In our analysis, we encountered that the wflow_sbm model, originally developed for small-scale applications, is less affected by the temporal discretisation error in small UK basins compared to the larger scale PCR-GLOBWB model. We recognize the significance of this finding, and we will emphasize it in the results section.

Additionally, we will identify and report the basins with large temporal discretisation errors. As suggested by the second reviewer we will implement a procedure that highlights catchments where the precipitation peak and streamflow peak coincide in the same time step.

*"The authors recognise, as in other large sample modelling studies, that there is a significant percentage of catchments for which models perform badly. In terms of considering the potential impacts of observation uncertainties, why is this not taken more seriously (it has also been ignored in all the recent machine learning studies). Is it not important to learn why that is the case (and no it is not all down to chalk catchments – and if your perceptual model already informed you that your models would not perform well on chalk catchments why did you make the applications? Needs justification, and not just because everyone else has included as many catchments as possible)"*

In our publication, we do indeed highlight instances where the model performance is unsatisfactory, leading to the conclusion that the models are not fit for purpose in these specific catchments. We believe it is crucial to present not only the success stories but also the complete picture of model performance when evaluating using large-sample datasets. By doing so, we provide a more accurate representation of the models capabilities and limitations.

Regarding the application of models to catchments where the perceptual model suggests potential poor performance, we acknowledge the importance of justification. While it is true that the model developers perceptual model might have indicated potential limitations for certain catchments, we believe there is value in exploring these cases further. By giving models the opportunity to fail, we allow for more comprehensive model evaluation and gain valuable insights into the reasons behind their poor performance. As pointed out earlier, these cases of seemingly poor model performance can be the most distinguishing and informative results. Our study, therefore, identifies potential areas for model improvement and guides future research efforts. We will perform further analysis to better understand the reasons why the models fail in particular catchments using a wider range of catchment descriptors.

*"L45. Correlation is not causality – it might actually be better to search for understanding at the local level."*

We agree with this statement and will adjust this in the revised manuscript by reflecting on the fact that correlation is not causation.

*"L49. Some important papers missing here - Beven and Smith JHE ASCE 2015; Beven HSJ 2016, PRSL 2019, Beven and Lane HP 2022, Beven et al HP 2022. L52. There were earlier papers – e.g. Liu et al JH 2009; Blazkova and Beven, WRR, 2009"*

We will extend the referenced literature based on these suggestions and will extend the literature further.

*"L114. Aggregated how? In a way consistent with the hydrological process descriptions?"*

The hydraulic parameters are upscaled using the method presented in Eilander et al. (2021). The parameter upscaling of the wflow_sbm model is based on the work by Imhoff et al. (2020) that used point-scale (pedo)transfer-function following the MPR technique by Samaniego et al. (2010). Parameters are aggregated from the original data resolution with upscaling operators determined by a constant mean and standard deviation across different scales. Fluxes and states are checked for consistency. See van Verseveld et al. (2022) for further information. We will better clarify the parameterization and aggregation of parameters in the revised manuscript.

*"L118/119. Return flow has a specific meaning in hillslope hydrology that is different? And surely water use and water demand are not included in this data set so are not used here?"*

We will refrain from using the term "return flow". Only a few large catchments in the Thames basin and parts of Scotland model water use and water demand by the PCR-GLOBWB model. This will be included in the Appendix of the revised manuscript.

*"L124. Respectively is the wrong way round!"*

Thank you, this will be adjusted in the revised manuscript.

*"L130. Does not require additional calibration? Why not – surely it could benefit????? Should you not just say it was applied without additional calibration. And why is a 30 year steady state based on average daily values an appropriate initial condition for the start of 2008? Would not seem appropriate for either baseflow dominated (chalk) catchments or flashier catchments? OK, at least 2008 was discarded so not too important."*

We agree that (any) model can be benefit from additional calibration. We implemented the model as intended by the model developers. Admittedly, the 30-year simulation period might be considered relatively short for reaching steady state conditions in certain hydrological systems, if not all. As mentioned, to address this issue we remove the year 2008 from our analysis. By doing so, we aimed to minimize any potential transient effects.

In our revised publication, we will explicitly state that the model has been calibrated by the developers. Furthermore, we will acknowledge the potential benefits of conducting additional calibration, which could lead to further improvement in the model results. By providing this clarification, we aim to ensure transparency and emphasize the importance of considering calibration choices in our study.

*"L140 why would you expect that lateral Ks should be much greater than vertical Ks? Are not most macropores in the near surface vertical. Is this an indication that the process representations are inadequate and sufficient to reject the model (e.g. subsurface celerities not being handled properly). And does a value of 100 already mean 100x or a factor of 1. Needs more discussion/clarification."*

We do not expect lateral Ks to be much greater than vertical Ks. This is a severe limitation of the wflow_sbm model that highlights deficiencies in process representations. Therefore, we do not expect an amplification factor to be necessary to correct the model. One notable concern is the reliance on a topographic gradient drive procedure, which may not fully account for the complexities of pressure

driven flow in hydrological system. Additionally, the models lack of preferential flow representation can lead to unrealistic parameter values that lack physical meaning.

In our study, we take the role of model users and aim to distinguish ourselves from the model developers by addressing these limitations. We will more explicitly describe the model calibration approach in the methods section.

*"L186/187. This is really unclear – averages of uncertainty bounds? Why do not these come simply from the rating curve uncertainties at each time step?"*

This is due to the data limitations that are available in the CAMELS-GB dataset. We accept this limitation as we promote the use of existing dataset to ensure community participation into implementing the suggested evaluation procedure in other studies.

*"L189/190. T-test?   But these are not independent values?"*

We applied a pair-wise T-test for dependent values. We will clarify this in the methods.

*"Figure 3.   Something seems wrong here.   On both plots the calibrated model has worse values than the default model for the values > 0"*

In Figure 3A, we can observe a slight overlap and lower performance among the wflow_sbm models. This can be attributed to two causes.

Firstly, the optimal parameter value obtained during calibration are often found to be similar to the default value for certain parameters. This suggests that the model calibration may not significantly improve the performance over the default configuration. The ever so slightly lower performance is due to the calibration process, where the model is calibrated using the average of individual water years over the entire calibration period. As a consequence, some parameters may end up with values that do not deviate significantly from their default settings.

Secondly, during the calibration process, we optimized the model for a single objective function (KGE-NP). While this approach allowed us to achieve a favourable performance with respect to the chosen objective, it may lead to variations in other objective functions. The results in Figure 3B demonstrate the duality of optimizing for a single objective function by how it affects the NSE objective function.

Given the importance of capturing multiple aspects of model performance, we found it necessary to include multiple objective functions in our analysis. By doing so, we could gain a more comprehensive understanding of the models capabilities and limitations, accounting for the trade-offs and interdependencies between different metrics.

*"L228.  The relevance of which is debatable?   Really???   The models are really poor for these sites  –  THAT is important - it is he relative values of just how bad are that is not so relevant."*

We recognize that the term "relevance" might have caused confusion. Our intention was to convey that models failing to outperform a simple benchmark, such as taking the mean of the observed flow, should be excluded from the analysis. In other words, we aim to exclude models that do not demonstrate a meaningful improvement over the most basic representation of observed flow.

We will be more explicit in the revised manuscript.

*"L230.  See comments above about outliers and disinformation."*

We will implement changes as mentioned in the comments above about outliers and disinformation.

*"L254. Not clear here – if you have taken average percentage uncertainties by flow class and multiplied by the flow then how is there such variation?"*

Here, we only refer to the uncertainty percentages without multiplication with flow. This will be clarified in the revised publication.

*"L265/266 is that not the inverse of what you started at the start of this paragraph?"*

Thank you, this will be corrected in the revised manuscript.

*"Section 4.1. See comments above about temporal variability, outliers and disinformation"*

We will implement similar changes as mentioned in the comments above about temporal variability, outliers and disinformation.

*"L294/295. But equifinality has been discussed for more than 20 years now – but does not get an explicit mention in the text anywhere?"*

We agree that this should be mentioned. We will discuss "equifinality" in the introduction and discussion supported by literature.

*"L300/301 So only take your best cases??? is it not more important to understand what is happening at these sites and allow for that understanding in what you use to predict? Might these be model invalidation sites (see Beven and Lane, HP 2022)"*

Indeed this is contradictory to what we mentioned earlier in this rebuttal. We will adjust this using the suggested concept and reference regarding model invalidation sites.

*"Section 4.3. See Beven HP 2023 benchmarking paper for an alternative view."*

We will incorporate the findings of the Beven (2023) commentary in the discussion section.

*"L346 experiment"*

Thank you for the technical corrections.

**References:**

Beven, K. (2023). Benchmarking hydrological models for an uncertain future. *Hydrological Processes*, *37*(5), e14882. https://doi.org/10.1002/hyp.14882

Beven, K., & Binley, A. (1992). The future of distributed models: Model calibration and uncertainty prediction. *Hydrological Processes*, *6*(3), 279–298. https://doi.org/10.1002/hyp.3360060305

Beven, K., & Freer, J. (2001). Equifinality, data assimilation, and uncertainty estimation in mechanistic modelling of complex environmental systems using the GLUE methodology. *Journal of Hydrology*, *249*(1), 11–29. https://doi.org/10.1016/S0022-1694(01)00421-8

Beven, K., & Lane, S. (2019). Invalidation of Models and Fitness-for-Purpose: A Rejectionist Approach. In C. Beisbart & N. J. Saam (Eds.), *Computer Simulation Validation: Fundamental Concepts, Methodological Frameworks, and Philosophical Perspectives* (pp. 145–171). Springer International Publishing. https://doi.org/10.1007/978-3-319-70766-2_6

Beven, K., & Smith, P. (2015). Concepts of Information Content and Likelihood in Parameter Calibration for Hydrological Simulation Models. *Journal of Hydrologic Engineering*, *20*(1), A4014010. https://doi.org/10.1061/(ASCE)HE.1943-5584.0000991

Clark, M. P., Vogel, R. M., Lamontagne, J. R., Mizukami, N., Knoben, W. J. M., Tang, G., Gharari, S., Freer, J. E., Whitfield, P. H., Shook, K. R., & Papalexiou, S. M. (2021). The Abuse of Popular Performance Metrics in Hydrologic Modeling. *Water Resources Research*, *57*(9), e2020WR029001. https://doi.org/10.1029/2020WR029001

Eilander, D., van Verseveld, W., Yamazaki, D., Weerts, A., Winsemius, H. C., & Ward, P. J. (2021). A hydrography upscaling method for scale-invariant parametrization of distributed hydrological models. *Hydrology and Earth System Sciences*, *25*(9), 5287–5313. https://doi.org/10.5194/hess-25-5287-2021

Fowler, K., Peel, M., Western, A., & Zhang, L. (2018). Improved Rainfall-Runoff Calibration for Drying Climate: Choice of Objective Function. *Water Resources Research*, *54*(5), 3392–3408. https://doi.org/10.1029/2017WR022466

Imhoff, R. O., van Verseveld, W. J., van Osnabrugge, B., & Weerts, A. H. (2020). Scaling Point-Scale (Pedo)transfer Functions to Seamless Large-Domain Parameter Estimates for High-Resolution Distributed Hydrologic Modeling: An Example for the Rhine River. *Water Resources Research*, *56*(4), e2019WR026807. https://doi.org/10.1029/2019WR026807

Kavetski, D., Kuczera, G., & Franks, S. W. (2006a). Bayesian analysis of input uncertainty in hydrological modeling: 1. Theory. *Water Resources Research*, *42*(3). https://doi.org/10.1029/2005WR004368

Kavetski, D., Kuczera, G., & Franks, S. W. (2006b). Bayesian analysis of input uncertainty in hydrological modeling: 2. Application. *Water Resources Research*, *42*(3). https://doi.org/10.1029/2005WR004376

Lamontagne, J. R., Barber, C. A., & Vogel, R. M. (2020). Improved Estimators of Model Performance Efficiency for Skewed Hydrologic Data. *Water Resources Research*, *56*(9), e2020WR027101. https://doi.org/10.1029/2020WR027101

McMillan, H. K., Westerberg, I. K., & Krueger, T. (2018). Hydrological data uncertainty and its implications. *WIREs Water*, *5*(6), e1319. https://doi.org/10.1002/wat2.1319

Samaniego, L., Kumar, R., & Attinger, S. (2010). Multiscale parameter regionalization of a grid-based hydrologic model at the mesoscale. *Water Resources Research*, *46*(5). https://doi.org/10.1029/2008WR007327

Shabestanipour, G., Brodeur, Z., Farmer, W. H., Steinschneider, S., Vogel, R. M., & Lamontagne, J. R. (2023). Stochastic Watershed Model Ensembles for Long-Range Planning: Verification and Validation. *Water Resources Research*, *59*(2), e2022WR032201. https://doi.org/10.1029/2022WR032201

van Verseveld, W. J., Weerts, A. H., Visser, M., Buitink, J., Imhoff, R. O., Boisgontier, H., Bouaziz, L., Eilander, D., Hegnauer, M., ten Velden, C., & Russell, B. (2022). Wflow_sbm v0.6.1, a spatially distributed hydrologic model: From global data to local applications. *Geoscientific Model Development Discussions*, 1–52. https://doi.org/10.5194/gmd-2022-182

Westerberg, I. K., Sikorska-Senoner, A. E., Viviroli, D., Vis, M., & Seibert, J. (2022). Hydrological model calibration with uncertain discharge data. *Hydrological Sciences Journal*, *67*(16), 2441–2456. https://doi.org/10.1080/02626667.2020.1735638

---

## Author Comment (AC2)

Reply to Review #2

We would like to start by thanking the anonymous reviewer for their constructive comments and insightful review. We believe strongly that the feedback will help strengthen the revised manuscript and thereby will be of value to the hydrological modelling community.

As mentioned in the reply to Review #1, we feel that the review may not fully capture the key message and central focus of our work, indicating that our framing requires significant improvement. In particular, we need to clarify the distinction between model developers and model users, a vital aspect for the proper framing of this study. As the authors of this publication, we belong to the model user category.

It is essential to clarify that our intention is not to overlook any structural issues in the inputs, models, or their applications. The central point of our study is: "How do users of models, whom are not themselves the developers of these models" have to interpret differences in model output in the light of discharge observation uncertainty. Our study provides these users of models with the workflow and statistical tools to make that decision. We make a clear distinction between model developers and model users, because in daily use of models this distinction is also present. We illustrate this workflow and its usefulness with the model runs done with the two models on the CAMELS-GB dataset. The models are employed in this study as intended by the developers to demonstrate inevitable short comings and strengths of these models. We recognize the need to address this framing issue throughout the publication after careful revision.

We are fully committed to addressing the specific concerns raised by the reviewer. In the following section, we will provide a point-by-point response to each concern and outline the steps we will take to address them adequately.

**General comments:**

*"A significant overhaul is needed in my opinion. Mainly, the title says "observation uncertainty" but precipitation is also an observation and its uncertainty is left out. The paper should have said "observed discharge uncertainty" because that is the only thing it deals with."*

In light of your feedback, We recognize that the publication needs to distinguish between the observation uncertainty and observed discharge uncertainty. To accurately represent our work, we will modify the title to 'observed discharge uncertainty,' as this encompasses the specific scope of our study. We will make sure to address this distinction throughout the revised manuscript.

**Specific comments:**

Abstract:

*What is temporal sampling and observation uncertainty?*

Indeed, these terms are ambiguous for readers and need clarification. We will improve the abstract by clarifying the terms and by using "discharge observation uncertainty" instead of "observation uncertainty". The term "temporal sampling uncertainty" will be clarified by stating that this refers to the uncertainty that stems from the sampling of the evaluation period for objective function calculation.

*"L2-3: While mentioning that comparison studies are invalid, it should also be mentioned that all models (and the inputs used) are invalid to begin with. No model incorporates true nature. In my opinion, the point is more about finding out models that are useful for a given purpose."*

We agree with the statement that models are flawed by nature and are only fit for a given research question (purpose). As mentioned in the review, if a model's purpose is to accurately predict river discharge, as in our study, it is relevant to include discharge observation uncertainty in model comparisons. Adjustments will be made to include this in the revised manuscript.

*L3-5: Regarding the problem of temporal sampling, same data is fed to all models. If some perform better than others then, isn't this what we are looking for?*

The aim is indeed to identify the best performing model. In hydrology, model often determined by implementing a singular objective function that quantifies the agreement between simulations and observations over a prolonged time period. As discussed in Clark et al. (2021), we must not overlook the relevance of temporal sampling when comparing hydrological models. For instance, Fowler et al. (2018) pointed out that equally weighting each year in the calibration timeseries emerges as the most robust strategy. This approach ensures that a given wet year's influence, that may have a significant impact on the KGE-NP score (this study), is balanced with the potential value contained in dry years, particularly in the context of a changing climate. Furthermore, the work of Lamontagne et al. (2020) highlights an essential aspect of certain objective functions, which might exhibit low bias but still manifest considerable variability between samples of the streamflow time series. This variability can stem from the skewness and periodicity of the data. It is this variability that should be adequately captured by the hydrological models. Therefore, we highlight the problem of temporal sampling as evaluation procedures that only assess model performance on the whole temporal period can obscure objective judgement.

*L10-13: Only two models are compared? I would have used may be 10 given how large the number of test catchments is. Gao et al. 2018 show many in their first table (both conceptual and physically-based). It would be interesting to see how the results change by taking more models.*

There is a wealth of hydrological models available of which only a small selection is presented in Gao et al. 2018. In this study, we demonstrate the use of an easy to implement methodology that is agnostic towards the selected models. Therefore, we feel that the addition of more models will not further exemplify the proposed method. We decided on using distributed models given their relevance for other research. Minimizing redundant model runs and optimises time spent on novel research (Jain et al. 2022.) In addition, we utilized the eWaterCycle platform for model experimentation as it ensures reproducibility of this research. At the time of writing the number of available models on this platform is still limited, but actively being expanded upon. The analyses in this manuscript may be re-run once more models are available on the platform.

*L11-13: For the inter-model case, please mention whether the models are calibrated or not.*

Thank you for this comment, we will rectify this in the revised manuscript.

**1. Introduction**

*L31: I am really really sorry for my nit picking but hydrologists were well aware of the challenging aspects of hydrological modeling long before 2018. It is a well-known problem. I think you can omit the citation.*

We don't view this as nit picking, we will remove the reference in the revised manuscript.

*L33-45: Very informative. Thanks.*

Thank you for this comment.

**2. Methodology**

*L71: Observation uncertainty is mentioned but temporal uncertainty isn't. Just add a few words for the sake of completeness.*

We will add a few words on temporal uncertainty in the revised manuscript for the sake of completeness.

*L94-99: Nicely summarized.*

Thank you for this comment.

*L103: Fine spatial resolution is used, but the problem of the daily temporal resolution is not treated. Small catchments (area < 1000 km2) have problems with time-of-concentrations. There, the peak precipitation and discharge take place at the same time step. Something that the model cannot solve and produces parameters that are unrealistic during calibration. The problem of a few values dominating the objective function is also the consequence of incorrect temporal resolution. At least in my experience. I have seen this problem for catchments of more than 4000 km2 size. And I have a sneaking suspicion that CAMELS-GB has smaller catchments inside it. A procedure has to be used that discards catchments where the precipitation peak and flow peak happen at the same time step, most of the times. Such a problem exists for larger catchment on daily time scales but to a much smaller degree. That is when a precipitation event happens near to the catchment mouth.*

Thank you for this insightful comment. We will identify and report the basins with large temporal discretisation errors as well as their catchment size. An additional analysis will be conducted following the mentioned procedure that highlights catchments where the precipitation peak and streamflow peak coincide in the same time step. The results will be critically reflected upon in the discussion chapter.

Your feedback reinforces the importance of elucidating temporal sampling uncertainty. For instance, a hydrological simulations inability to capture peak flow might still yield a high objective function value when evaluated across an extended period. The potential penalization for missing peak flow can be mitigated by adequately capturing other components, such as baseflow. This underscores the intricate nature of the temporal sampling challenge, which we acknowledge in the study.

*L105-106: It is not mentioned why only two models were chosen. I don't understand why legacy gave us two models only. In GB, Keith Beven has, for sure, used others.*

This is an ill-used term that will be removed from the revised manuscript. Originally legacy, referred to the relevance for evaluating hydrological models in the context of "hyper resolution modelling" (e.g, Wood et al., 2011; Beven and Cloke, 2012; Bierkens et al., 2015).

*L127: Yes, they will most probably lead to different conclusions. Hence, the recommendation of more than two models.*

The term in question has been misapplied and will be removed from the revised manuscript. Initially, it was employed to denote its relevance in assessing hydrological models within the framework of "hyper resolution modeling".

*L129-131: Please elaborate as to why additional calibration is not needed. I do not understand. Is it so that model parameters are somehow known already? Comparing uncalibrated models to calibrated ones is unfair in my opinion.*

It's important to reframe the study to underscore that the authors are model users, not developers. We applied the models in accordance with the developers' intended use, meaning they were already calibrated and ready for direct application. Thus, evaluating the advantages of further calibration remains relevant. Although this comparison might appear biased, our curiosity lies in assessing

whether enhanced model performance remains meaningful when considering discharge observation uncertainty.

*L145-151: Interesting. I am glad that this study relies so much on other's work and doesn't try to reinvent the wheel.*

We agree with the notion that adopting or relying on other's qualitative work should be done when possible.

*L161-174: Here, I have a major problem with this study. The problem being that input uncertainty is not taken into account. Normally, observation locations are not enough to capture the point of the maximum precipitation which consequently leads to underestimation (in some cases overestimation) of the precipitation volume. This problem was demonstrated recently in Bárdossy and Anwar 2023. From the methodology explained here, I do not see any mention/treatment of this major problem till now.*

To address this significant concern, we will implement two key changes. Firstly, we will reframe the study to focus on discharge observation uncertainty, which is more accurate in capturing the essence of our investigation. Secondly, we acknowledge the necessity of addressing the aforementioned issue explicitly. We will introduce a discussion on this major problem in both the introduction and methodology sections. This will include a reflection on the challenges of performing demonstrations such as in Bárdossy and Anwar (2023) on a large-sample of catchments. Additionally, a dedicated section in the discussion will encompass a comprehensive reflection on all sources of input uncertainty. This will emphasize that our study constitutes just a fraction of a broader challenge, in which input uncertainty plays a substantial role.

*Also, I saw in the CAMELS-GB paper that they give a mean value of precipitation over the catchment. This is problematic, if used for a distributed model.*

Our study used distributed precipitation, temperature, and potential evapotranspiration fields as inputs. The CAMELS-GB paper used the same data to derive a single mean value per catchment.

*However, dealing with input, model and discharge uncertainty is an ill-posed problem and doesn't seem to have any acceptable solution, as far as I know. A study that deals with observation uncertainty and leaves out precipitation will, in my humble opinion, lead to incorrect/invalid conclusions.*

*One could argue that all models are presented with the same input, and therefore it is not much of a problem. But then, why consider observation uncertainty as all models are evaluated based on the same discharge?*

The point you raise about the uniformity of input might suggest that discharge observation uncertainty is less significant. However, it's important to note that while meteorological inputs might be similar, parameterization uncertainties and inherent model uncertainties vary between models. This leads to different propagation of uncertainties. The distinction in uncertainty propagation necessitates the consideration of discharge observation uncertainty, even when evaluations are based on the same observations.

*Slightly off topic. I think the readers would benefit if the temporal uncertainty methods are summarized like other topics previously. Using words like bootstrapping and jackknifing are not so helpful. After all, it is what the study is about.*

A short summary that does not rely on abstract terms would definitely benefit the reader. This will be adjusted in the revised manuscript.

*L175-181: Don't all models struggle with the upper 5% of the distribution? I find it disconcerting that such an important detail is left out and is only mentioned now. These are the flows that cause actual problems; this is where major timing and volume problems exist and these are the time steps where the squared error dominates the objective function generally. I understand that not enough data was available but leaving the good stuff out is akin to ignoring the major problem at hand. Such details should be mentioned in the abstract as many are interested in the upper 5%. The low flows, I can forgive as they are contaminated by wastewater flowing in to the river which may or may not be originating, in terms of source, from the same catchment.*

This is a very valid point, we will rectify this by extending the analysis to include the limited set of catchments simulations that contain estimates for the upper percentiles of the uncertainty distribution. We appreciate your perspective on the significance of this detail, especially given that these flows often pose real-world challenges. We agree that not addressing this aspect can be misleading and downplays a major issue.

Furthermore, we acknowledge the potential for this analysis to underscore both the importance of the temporal sampling issue and the relevance of considering the upper 5% in hydrological modelling assessments.

*L183: What is model A and B in figure 2A? Are 1A and 2A showing the same event? There is no value of discharge on the y-axis. How is one supposed to tell, say, whether there was an actual peak when model B also shows a peak or just that model B spontaneously rose to a high value? And given that A is not as reactive as B, my guess is that something is very wrong with A.*

To clarify, this figure was intentionally designed as an extreme example to illustrate our methodology, without cherry-picking specific events. We do recognize the need for better contextualization. In response to your feedback, we will enhance the figure's caption by explicitly stating that the illustration is an extreme case intended for methodological exemplification. This clarification will help readers better understand the purpose and context of the figure.

**3. Results**

*L214-215: I find such a comparison to be meaningless. wflow_sbm default has some quasi-arbitrary parameters. These could have performed very good or very bad. In my opinion, if one has to take one single model, the calibrated model is the one because it the best we could do, assuming that the validation also shows improvement. Also, only one parameter was optimized. Which I find strange. If there is access to a supercomputer then, why not all (that may be optimized)? It would be interesting to see. I have no attachment with wflow_sbm or PCR-GLOBWB, but some sort of parameter optimization should also be done for this. At the end, it could be what the authors point out about how it routes flow. We would only know if optimization is carried out.*

As mentioned earlier, this comes down to the distinction between model users and developers. We employed the models as intended by the model developers. The models are readily calibrated and already optimized on HPC infrastructure. The reason for optimizing only a single parameter for the wflow_sbm model is that this was identified as an effective parameter for calibration that increases baseflow and decrease peak flow (Imhoff et al., 2020; Aerts et al., 2022). We find it therefore of interest to see what this additional calibration step entails from a model user perspective.

*L253-255: The relative low flow uncertainty could be higher due to wastewater being introduced into the streams as I mentioned earlier. And could also be due the presence of karst that is mentioned by the authors.*

Thank you for this insight, we will search for wastewater estimates to confirm this notion. If not available, we will investigate the relationship between human influence and low flow uncertainty as

both are available in the CAMELS-GB dataset. The results of this additional analysis will be reported in the revised manuscript.

*L258-267: Here, I would like to stress again that without considering input uncertainty, the conclusions are incorrect as the results are (heavily) influenced by input (precipitation). And the authors should acknowledge this.*

We recognize the significant influence of input, particularly precipitation, on our results. To address this concern, we will include a dedicated section discussing input uncertainty and its broader implications. This discussion will underscore the interconnected nature of discharge observation uncertainty within a larger context.

**4. Discussion:**

*L269-274: The authors finally mention the other sources of uncertainty this deep in the text. The reason why this problem is over-looked (by hydrologists that know about this problem) is that evaluating uncertainties is not trivial and requires much more data (which Coxon (2015) had the luxury of to some extent) computational power, and many assumptions (that likely remain unfulfilled or cannot be verified to hold). Working with uncertain data has been tried before but all end up at the same point i.e., if uncertainties have to be handled then the proper way is to take all types in to account simultaneously. This is a major problem. Uncertainty bounds of any variable are calculated, normally, using Gaussian-dependence. For precipitation, for example, advection and convection exists. Something that interpolation schemes cannot capture by considering only a subset of points in the catchment. Also, they are non-Gaussian fields. Radar shows some structure of the precipitation field but is also limited in its capabilities when it comes to precipitation volumes and is not always better than using gauge data.*

Thank you for this view on working with uncertain data, especially in the context of precipitation. The suggested literature by Bárdossy and Anwar clearly demonstrates to us how this impacts rainfall-runoff models. While acknowledging the limitations of current uncertainty estimation methods, we recognize the value of data sources such as CAMELS-GB in exploring uncertainties. Despite relying on assumptions like Gaussian mixture models, these sources offer valuable insights into the broader uncertainty landscape. In our discussion chapter, we will provide a more comprehensive reflection on these limitations, as well as the importance of incorporating all relevant sources of input uncertainty.

**References:**

Aerts, J. P. M., Hut, R. W., van de Giesen, N. C., Drost, N., van Verseveld, W. J., Weerts, A. H., & Hazenberg, P. (2022). Large-sample assessment of varying spatial resolution on the streamflow estimates of the wflow_sbm hydrological model. *Hydrology and Earth System Sciences*, *26*(16), 4407–4430. https://doi.org/10.5194/hess-26-4407-2022

Bárdossy, A., & Anwar, F. (2023). Why do our rainfall–runoff models keep underestimating the peak flows? *Hydrology and Earth System Sciences*, *27*(10), 1987–2000. https://doi.org/10.5194/hess-27-1987-2023

Beven, K. J., & Cloke, H. L. (2012). Comment on "Hyperresolution global land surface modeling: Meeting a grand challenge for monitoring Earth's terrestrial water" by Eric F. Wood et al. *Water Resources Research*, *48*(1). https://doi.org/10.1029/2011WR010982

Bierkens, M. F. P., Bell, V. A., Burek, P., Chaney, N., Condon, L. E., David, C. H., de Roo, A., Döll, P., Drost, N., Famiglietti, J. S., Flörke, M., Gochis, D. J., Houser, P., Hut, R., Keune, J., Kollet, S., Maxwell, R. M., Reager, J. T., Samaniego, L., … Wood, E. F. (2015). Hyper-resolution global hydrological modelling: What is next? *Hydrological Processes*, *29*(2), 310–320. https://doi.org/10.1002/hyp.10391

Clark, M. P., Vogel, R. M., Lamontagne, J. R., Mizukami, N., Knoben, W. J. M., Tang, G., Gharari, S., Freer, J. E., Whitfield, P. H., Shook, K. R., & Papalexiou, S. M. (2021). The Abuse of Popular Performance Metrics in Hydrologic Modeling. *Water Resources Research*, *57*(9), e2020WR029001. https://doi.org/10.1029/2020WR029001

Imhoff, R. O., van Verseveld, W. J., van Osnabrugge, B., & Weerts, A. H. (2020). Scaling Point-Scale (Pedo)transfer Functions to Seamless Large-Domain Parameter Estimates for High-Resolution Distributed Hydrologic Modeling: An Example for the Rhine River. *Water Resources Research*, *56*(4), e2019WR026807. https://doi.org/10.1029/2019WR026807

Jain, S., Mindlin, J., Koren, G., Gulizia, C., Steadman, C., Langendijk, G. S., Osman, M., Abid, M. A., Rao, Y., & Rabanal, V. (2022). Are We at Risk of Losing the Current Generation of Climate Researchers to Data Science? *AGU Advances*, *3*(4), e2022AV000676. https://doi.org/10.1029/2022AV000676

Lamontagne, J. R., Barber, C. A., & Vogel, R. M. (2020). Improved Estimators of Model Performance Efficiency for Skewed Hydrologic Data. *Water Resources Research*, *56*(9), e2020WR027101. https://doi.org/10.1029/2020WR027101

Wood, E. F., Roundy, J. K., Troy, T. J., van Beek, L. P. H., Bierkens, M. F. P., Blyth, E., de Roo, A., Döll, P., Ek, M., Famiglietti, J., Gochis, D., van de Giesen, N., Houser, P., Jaffé, P. R., Kollet, S., Lehner, B., Lettenmaier, D. P., Peters-Lidard, C., Sivapalan, M., … Whitehead, P. (2011). Hyperresolution global land surface modeling: Meeting a grand challenge for monitoring Earth's terrestrial water. *Water Resources Research*, *47*(5). https://doi.org/10.1029/2010WR010090

---

## Author Response (AR1)

Point-by-point response Review #1

Dear Prof. Beven,

We wish to express our gratitude for your comprehensive and insightful review of our manuscript. Your feedback has been invaluable, serving not only as a guide for substantial revisions but also as a catalyst for enhancing the manuscript's overall clarity, depth, and scientific value. Below, we first detail the extensive revisions made followed by a point-by-point response to your comments.

**Clarification of the study's aim, positioning, and structure:**
- The title is revised to "On the importance of discharge observation uncertainty when interpreting hydrological model performance" and the publication only refers to "observation uncertainty" as discharge observation uncertainty. In addition, the words "outliers" and "return flow" are removed from the publication.
- The study's aim is clarified by clearly stating that this study is performed to provide model users with an assessment of the effect of omitting discharge observation uncertainty while interpreting model performance differences. *L1-9, L83-85*.
- To strengthen this point, 3 use cases are introduces that also improve the structure of the publication. *L103-115*
- The publication states and the results emphasize that discharge observation uncertainty and input uncertainties need not be included in model calibration and model evaluation efforts. *L20-23, L373-378, L418-420, L452-455*
- The various sources of uncertainty within hydrological modelling and the concept of equifinality are now extensively discussed in the introduction (*L26-59)* and discussion (*L373-379)*.
- The relevance of conducting a temporal sampling of the simulation and observation pairs for model performance interpretation is clarified. *L351-354, L399-413.*
- Input uncertainty is mentioned in the introduction and the importance of including input uncertainties is stressed in the discussion section.
- A section in the discussion reflects on the practical implications of the results for model users. L415-425.222
- A section is added in the discussion that reflects on the limitations of this study. *L426-435*

**Methodological Revisions:**
- Catchments are excluded from the analysis that indicate temporal discretization errors in small catchments due to peak precipitation and peak discharge occurring at the same time step. This is done by calculating the cross-correlation between observed discharge and precipitation for a range of lag times. Catchments that predominantly show less than 1 day of lag between observed discharge and precipitation are excluded.
- A new use case is introduced that compares the model structural uncertainty of six additional conceptual models. The model structural uncertainty is expressed in the maximum model discharge simulation difference and is subsequently compared to the discharge observation uncertainty estimates. *L273-296, L329-368*

**General revisions:**
- Improvements to overall written text.
- We extended the body of literature.
- We extended conclusions.

- Improvement of Figure 2 by selecting a more relatable example.
- Addition of model structure use case to Figures 3,4 and Table 2.
- Addition of Figure 6, comparing six individual conceptual hydrological models.
- Addition of Figure 7, indicating the effects of temporal sampling uncertainty for 6 conceptual hydrological models.

Point-by-point response:
- *"L258-267: Here, I would like to stress again that without considering input uncertainty, the conclusions are incorrect as the results are (heavily) influenced by input (precipitation). And the authors should acknowledge this."*
  *"Input uncertainties should not be ignored, as has been done here, since they can be an important source of disinformation in calibration data sets."*

- We mention the various uncertainty sources, there potential implications for hydrological modelling, and the equifinality concept at the beginning of the introduction and in the discussion section. The publication states and the results emphasize that discharge observation uncertainty and input uncertainties need not be included in model calibration and model evaluation efforts. *L20-23, L373-378, L418-420, L452-455*

- *"The authors refer to "outliers", but the authors do not differentiate between whether outliers might be the most important events in distinguishing models (e.g. Singh and Bardossy, AWR 2012), or whether they might in some cases introduce disinformation into the calibration process (e.g. the runoff coefficients greater than 1 of Beven and Smith, JHE ASCE 2015; Beven, PRSL 2019; Beven et al., HP 2022, 2023)."*

  We removed the term "outliers" as well as the term "return flow". In addition, we have included the suggested publications and mention the invalidation sites concept.

- *"Is the temporal sampling issue (in terms of different sampling periods) really relevant? Should we not give models the maximum chance to fail over extremes (after assessing for possible disinformation) by using as much data as possible (Shen et al., WRR 2022 also recently suggested this as the most robust calibration for use in prediction)."*

  We have improved the justification and clarification of the temporal sampling uncertainty analysis. *L351-354, L399-413.*

- *"There are, however, other temporal sampling issues in terms of the discretisation error of using daily time steps on some rather small UK basins. This really should be taken into account in that for some events it might be more significant than the rating curve error depending if an observed peak falls on one day or another relative to the model prediction."*

  Catchments are excluded from the analysis that indicate temporal discretization errors in small catchments due to peak precipitation and peak discharge occurring at the same time step. This is done by calculating the cross-correlation between observed discharge and precipitation for a range of lag times. Catchments that predominantly show less than 1 day of lag between observed discharge and precipitation are excluded. *L124-230*

- *"The authors recognise, as in other large sample modelling studies, that there is a significant percentage of catchments for which models perform badly. In terms of considering the potential impacts of observation uncertainties, why is this not taken more seriously (it has also been ignored in all the recent machine learning studies). Is it not important to learn why that is the case (and no it is not all down to chalk catchments – and if your perceptual model already informed you that your models would not perform well on chalk catchments why did you make the applications? Needs justification, and not just because everyone else has included as many catchments as possible)"*

  The results section is restructured and we now refrain from discussing individual model performance as this is outside of the scope of the study. Nonetheless, we mention this as a limitation in the added limitations section in the discussion.

- "L45. Correlation is not causality – it might actually be better to search for understanding at the local level."

  We agree and removed this statement.

- *"L130. Does not require additional calibration? Why not – surely it could benefit????? Should you not just say it was applied without additional calibration. And why is a 30 year steady state based on average daily values an appropriate initial condition for the start of 2008? Would not seem appropriate for either baseflow dominated (chalk) catchments or flashier catchments? OK, at least 2008 was discarded so not too important."*

  We reframed this statement making it clear that this is common practice by the model developers.

- *"L186/187. This is really unclear – averages of uncertainty bounds? Why do not these come simply from the rating curve uncertainties at each time step?"*
  *"L189/190. T-test? But these are not independent values?"*

  Both are clarified in the methodology and the absence of rating curve uncertainties is mentioned in the limitations.

- *L228. The relevance of which is debatable? Really??? The models are really poor for these sites – THAT is important - it is he relative values of just how bad are that is not so relevant."*

  Very ill-posed and removed from the publication.

The detailed revisions outlined above, undertaken in direct response to your feedback, have improved the manuscript's clarity, depth, and scientific value. We believe these revisions have comprehensively addressed your concerns, contributing to a manuscript that offers valuable insights and advancements to the hydrological modeling community.

On behave of the co-authors,

Jerom Aerts

Point-by-point response Review #2

Dear Reviewer,

We extend our sincerest thanks for your thorough review and the valuable feedback provided on our manuscript. Your detailed comments have prompted us to undertake a comprehensive revision process, significantly improving the manuscript's clarity and scientific value. In what follows, we detail the revisions made in response to each of your comments.

**Clarification of the study's aim, positioning, and structure:**
- The title is revised to "On the importance of discharge observation uncertainty when interpreting hydrological model performance" and the publication only refers to "observation uncertainty" as discharge observation uncertainty. In addition, the words "outliers" and "return flow" are removed from the publication.
- The study's aim is clarified by clearly stating that this study is performed to provide model users with an assessment of the effect of omitting discharge observation uncertainty while interpreting model performance differences. *L1-9, L83-85*.
- To strengthen this point, 3 use cases are introduces that also improve the structure of the publication. *L103-115*
- The publication states and the results emphasize that discharge observation uncertainty and input uncertainties need not be included in model calibration and model evaluation efforts. *L20-23, L373-378, L418-420, L452-455*
- The various sources of uncertainty within hydrological modelling and the concept of equifinality are now extensively discussed in the introduction (*L26-59)* and discussion (*L373-379)*.
- The relevance of conducting a temporal sampling of the simulation and observation pairs for model performance interpretation is clarified. *L351-354, L399-413.*
- Input uncertainty is mentioned in the introduction and the importance of including input uncertainties is stressed in the discussion section.
- A section in the discussion reflects on the practical implications of the results for model users. L415-425.222
- A section is added in the discussion that reflects on the limitations of this study. *L426-435*

**Methodological Revisions:**
- Catchments are excluded from the analysis that indicate temporal discretization errors in small catchments due to peak precipitation and peak discharge occurring at the same time step. This is done by calculating the cross-correlation between observed discharge and precipitation for a range of lag times. Catchments that predominantly show less than 1 day of lag between observed discharge and precipitation are excluded.
- A new use case is introduced that compares the model structural uncertainty of six additional conceptual models. The model structural uncertainty is expressed in the maximum model discharge simulation difference and is subsequently compared to the discharge observation uncertainty estimates. *L273-296, L329-368*

**General revisions:**
- Improvements to overall written text.
- We extended the body of literature.
- We extended conclusions.

- Improvement of Figure 2 by selecting a more relatable example.
- Addition of model structure use case to Figures 3,4 and Table 2.
- Addition of Figure 6, comparing six individual conceptual hydrological models.
- Addition of Figure 7, indicating the effects of temporal sampling uncertainty for 6 conceptual hydrological models.

Point-by-point response:
- *"A significant overhaul is needed in my opinion. Mainly, the title says "observation uncertainty" but precipitation is also an observation and its uncertainty is left out. The paper should have said "observed discharge uncertainty" because that is the only thing it deals with."*

  The title has been adjusted.

- *"L2-3: While mentioning that comparison studies are invalid, it should also be mentioned that all models (and the inputs used) are invalid to begin with. No model incorporates true nature. In my opinion, the point is more about finding out models that are useful for a given purpose."*

  We have removed this statement from the publication as we agree with your comment.

- *"L3-5: Regarding the problem of temporal sampling, same data is fed to all models. If some perform better than others then, isn't this what we are looking for?"*

  We have improved the justification and clarification of the temporal sampling uncertainty analysis. *L351-354, L399-413.*

- *"L10-13: Only two models are compared? I would have used may be 10 given how large the number of test catchments is. Gao et al. 2018 show many in their first table (both conceptual and physically-based). It would be interesting to see how the results change by taking more models."*

  We have added 6 conceptual hydrological models to the analysis.

- *"L11-13: For the inter-model case, please mention whether the models are calibrated or not."*

  We have removed the use of intra- and inter-model cases and substituted these for use cases for clarification and better structure.

- *"L103: Fine spatial resolution is used, but the problem of the daily temporal resolution is not treated. Small catchments (area < 1000 km2) have problems with time-of-concentrations. There, the peak precipitation and discharge take place at the same time step. Something that the model cannot solve and produces parameters that are unrealistic during calibration. The problem of a few values dominating the objective function is also the consequence of incorrect temporal resolution. At least in my experience. I have seen this problem for catchments of more than 4000 km2 size. And I have a sneaking suspicion that CAMELS-GB has smaller catchments inside it. A procedure has to be used that discards catchments where the precipitation peak and flow peak happen at the same time step,*

*most of the times. Such a problem exists for larger catchment on daily time scales but to a much smaller degree. That is when a precipitation event happens near to the catchment mouth."*

Catchments are excluded from the analysis that indicate temporal discretization errors in small catchments due to peak precipitation and peak discharge occurring at the same time step. This is done by calculating the cross-correlation between observed discharge and precipitation for a range of lag times. Catchments that predominantly show less than 1 day of lag between observed discharge and precipitation are excluded. *L124-230*

- *"L129-131: Please elaborate as to why additional calibration is not needed. I do not understand. Is it so that model parameters are somehow known already? Comparing uncalibrated models to calibrated ones is unfair in my opinion."*

The description of calibration and justification is now better described in the methods section.

- *"L161-174: Here, I have a major problem with this study. The problem being that input uncertainty is not taken into account. Normally, observation locations are not enough to capture the point of the maximum precipitation which consequently leads to underestimation (in some cases overestimation) of the precipitation volume. This problem was demonstrated recently in Bárdossy and Anwar 2023. From the methodology explained here, I do not see any mention/treatment of this major problem till now."*

The various sources of uncertainty within hydrological modelling and the concept of equifinality are now extensively discussed in the introduction (*L26-59*) and discussion (*L373-379*). In addition, input uncertainty is mentioned in the introduction and the importance of including input uncertainties is stressed in the discussion section.

- *"L175-181: Don't all models struggle with the upper 5% of the distribution? I find it disconcerting that such an important detail is left out and is only mentioned now. These are the flows that cause actual problems; this is where major timing and volume problems exist and these are the time steps where the squared error dominates the objective function generally. I understand that not enough data was available but leaving the good stuff out is akin to ignoring the major problem at hand. Such details should be mentioned in the abstract as many are interested in the upper 5%. The low flows, I can forgive as they are contaminated by wastewater flowing in to the river which may or may not be originating, in terms of source, from the same catchment."*

We now clearly state throughout the publication that the lower and upper 5% of flow are not considered in the analysis. In addition, this is mentioned in the added limitations section in the discussion. *L426-435*

- *"L183: What is model A and B in figure 2A? Are 1A and 2A showing the same event? There is no value of discharge on the y-axis. How is one supposed to tell, say, whether there was an actual peak when model B also shows a peak or just that model B spontaneously rose to a high value? And given that A is not as reactive as B, my guess is that something is very wrong with A."*

We have selected a more relatable example hydrograph for this exemplary figure in the methodology.

The results section is restructured and we now refrain from discussing individual model performance as this is outside of the scope of the study.

The detailed revisions outlined above, undertaken in direct response to your feedback, have improved the manuscript's clarity, depth, and scientific value. We believe these revisions have comprehensively addressed your concerns, contributing to a manuscript that offers valuable insights and advancements to the hydrological modeling community.

On behave of the co-authors,

Jerom Aerts

---

## Referee Report (RR1)

General comments:

I am glad to see a revised version whose scope is reduced and all my comments are addressed. The paper is well organized but I found many typos. Addition of the conceptual models and their performance CDFs is very informative and more practical as not everyone uses physically-based models. The impressive performance of GR4J, given the few number of parameters, is something I did not expect. I only have minor comments now that are stated below, the authors can incorporate them quickly.

Specific comments:

L239-240: Are the discharges coming from calibrated models? It is not clear. I am assuming they are.

L290-296: Here, the conclusion drawn about differences being more evident in north and south as compared the center are subjective (weak). It would be simpler to just divide them via arbitrary (exact angles need to be determined) lines that divide the area into three portions and then comparing either the means or medians of KGE-NP differences. If this point is the focus of the study, then please be thorough. I don't find it relevant.

L298-305: Yes, but showing relative differences is not a good idea. River width is not shaped by low flows. During low flows, a slight fluctuation in depth can easily translate to above 50% flow increase/decrease. Sedimentation may result in an increase of a few centimeters, depending on the situation. The flow volume that passes through a cross-section is important. Here, another paragraph can be added that translates the relative differences to absolute flow volumes. I think, that would show a different story.

L437-455: The abstract and conclusions deal with the problem stated in the title now. The only problem that remains is getting uncertainty bounds on observed discharges, elsewhere in the world but the results do shed light on what the situation might look like.

Cheers

---

## Author Response (AR2)

Dear reviewer,

Thank you for considering this publication after major revision and providing additional feedback. Here, we will provide a point-by-point reply to your latest comments.

**Point-by-point reply:**

General comments:
I am glad to see a revised version whose scope is reduced and all my comments are addressed. The paper is well organized but I found many typos. Addition of the conceptual models and their performance CDFs is very informative and more practical as not everyone uses physically-based models. The impressive performance of GR4J, given the few number of parameters, is something I did not expect. I only have minor comments now that are stated below, the authors can incorporate them quickly.

*Thank you for your feedback. We are pleased to hear that the revised manuscript, with its reduced scope and incorporation of your comments, is now well-organized and informative. We have addressed the typos and improved the grammar throughout the manuscript based on your valuable feedback without altering the core content of the originally revised version.*

Specific comments:
L239-240: Are the discharges coming from calibrated models? It is not clear. I am assuming they are.

*Yes, the discharges are indeed derived from calibrated models. We have clarified this in the revised manuscript to remove any ambiguity regarding their origin.*

L290-296: Here, the conclusion drawn about differences being more evident in north and south as compared the center are subjective (weak). It would be simpler to just divide them via arbitrary (exact angles need to be determined) lines that divide the area into three portions and then comparing either the means or medians of KGE-NP differences. If this point is the focus of the study, then please be thorough. I don't find it relevant.

*We agree that the initial statements were subjective and not well-supported. We have removed the statements regarding visual trends in the results, as they were not essential to the manuscript's focus. Instead, we retained a brief analysis to complement the large-sample catchment assessments, acknowledging that while the results may not be conclusive, they provide additional context.*

L298-305: Yes, but showing relative differences is not a good idea. River width is not shaped by low flows. During low flows, a slight fluctuation in depth can easily translate to above 50% flow increase/decrease. Sedimentation may result in an increase of a few centimeters, depending on the situation. The flow volume that passes through a cross-section is important. Here, another paragraph can be added that translates the relative differences to absolute flow volumes. I think, that would showa different story.

*Thank you for raising this valid point. We address this issue in the limitations of the manuscript. The goal of the original plot showing uncertainty percentages is to effectively*

*communicate that these percentages can be substantial, highlighting the relative uncertainty across different flow categories (low, average, high). We agree that the flow categories are not equally sensitive to the physical phenomena that are the cause of discharge observation uncertainty. Using relative values (percentages) allows us to illustrate how uncertainty varies across different flow regimes, which is crucial given the wide range of flow magnitudes (e.g., from 1 m³/s to 100 m³/s) among the catchments. Absolute values result in boxplots with an extensive range, skewing the interpretation due to the large variability in flow volumes.*

*Using the three flow categories, we show how uncertainty affects low flows differently than high flows. This approach helps model users understand and interpret the results in the context most relevant to their applications. Given the variation in flow conditions, it is up to the model user to apply this information according to their needs. Therefore, we believe that relative values provide a clearer comparison and better convey the practical implications of uncertainty in our study.*

L437-455: The abstract and conclusions deal with the problem stated in the title now. The only problem that remains is getting uncertainty bounds on observed discharges, elsewhere in the world but the results do shed light on what the situation might look like.

*Thank you for your feedback. We are pleased that the abstract and conclusions now align with the problem stated in the title. We acknowledge the challenge of obtaining uncertainty bounds on observed discharges globally, which remains a significant issue. This broader challenge should be addressed by the scientific community as we expect discharge observation uncertainty to be substantial in regions where monitoring equipment might be of lesser quality and less stringent maintenance protocols.*

*While our study provides valuable insights into potential uncertainty scenarios for the specific use case in Great Britain, we recognize that extending these findings to a global context presents additional complexities. The results shed light on what the situation might look like, but further research is needed to address uncertainty bounds on observed discharges worldwide.*

*We appreciate your understanding and will highlight this in the outlook of  the revised manuscript to ensure clarity regarding the scope and applicability of our findings.*

**Main Changes:**
Manuscript: Grammar and spelling of the manuscript. A track-changes document is attached that provides all grammar and spelling corrections.

L239-240: Clarification of the type of model run, calibrated.

L286-293: Removal of text stating subjective spatial trends in the result.

L429-434: Reflection on the limitations of using relative values for expressing discharge observation uncertainty rather than absolute values.

L464-468: Addition of an outlook on the larger challenge of global discharge observation uncertainty estimates and its effect on water resources management.

---

## Author Response (AR3)

Dear Editor,

Thank you for considering our publication after major revision and for providing additional feedback. We appreciate your thorough review and the opportunity to improve our work further. Below, we provide a point-by-point response to your comments.

Figure 1 caption: consider to rephrase ", in green the model experiment inputs are shown, in red the models, and in grey the analysis components." to " The model experiment inputs are shown in green, the models in red, and the analysis components in grey."
Change capital A, B, and C letters in the flowchart to (a), (b), and (c) to be consistent.

*We appreciate your suggestion to rephrase the caption. We have revised the caption to: "The model experiment inputs are shown in green, the models in red, and the analysis components in grey." We have also updated the labels in the flowchart, changing the capital A, B, and C letters to A, b, and c to maintain consistency with the labeling convention used elsewhere in the publication. In addition, we have adjusted the font to improve text clarity.*

Lines 251-256 and Figure 2: Lines 251-256 need more clarity to what exactly you are presenting in Figure 2 and how you derived the so-called uncertainties. For example, you state "The discharge observation uncertainty estimates of the CAMELS-GB dataset are processed by averaging the upper and lower bounds of uncertainty estimates per flow percentile (5, 25, 75, 95)." Did you average all the four flow percentiles? How did you get +20% and -15%?

*Thank you for pointing out the need for more clarity in explaining Figure 2 and the derivation of the uncertainty estimates. We have revised the text to provide a more detailed and precise explanation of the process. Specifically, we clarified that:*

- *The absolute error, model difference, between calibrated model simulations and observed discharges for each flow category and catchment is calculated first, which is now explicitly visualized in Figure 2a (blue line).*
- *The upper and lower uncertainty bounds were taken from the CAMELS-GB dataset. We revised the text to state how these bounds were derived clearly: for example, the upper uncertainty percentages at flow category boundaries correspond to 25% and 15%, respectively, which we averaged to obtain an upper uncertainty bound of 20% for the low flow category. This is now clearly shown by the orange and red lines in Figure 2b.*
- *We also clarified the derivation of the overall observation uncertainty percentage, which is now explained as the average of the absolute values of the upper and lower bounds, resulting in a 17.5% uncertainty.*

The legend used in (b) and (c) can be confusing as they use the same green line and the same name to refer to two different things. Please consider updating one of them so one can differentiate the two lines that are currently both shown in green as 'Observation Uncertainty (17.5%). The way the name is given to a time series is already unclear. I guess the green line in (b) refers to 'the mean observation error' and in (c), 'the observed discharge with added uncertainties'.

*We acknowledge your concern regarding the potential confusion caused by using the same green line and label in Figures 2b and c. To address this, we have made several revisions to improve the example. These include changing the legend colors. Updating the time series data to another catchment ensures that the presentation is clearer and more consistent. These changes better*

*illustrate the method. process. In addition, We have clearly stated in the text that observation uncertainty refers to " the portion of the observed discharge attributed to uncertainty"*

Figure 3b Here, the optimized wflow_sbm model performs better in 75% of the catchments than the PCR-GLOBWB model. Both models demonstrate poor results for approximately 25% of the evaluated catchments (<0.40 KGE-NP).
Can you please double check the number 25%? It does not look right to me.

*We have adjusted the text to: "between approximately 18 % and 24 %".*

Table 2
Why do you need the last column to show total instances of 299? It is just repeated three times in the table. You already stated 299 catchments in the table caption.

*We removed the mention of the total instances in Table 2.*

Line 356 The HBV-96 and XINANJIANG models that most closely resemble each other based on the number of stores, process descriptions, and parameters
This is a wrong statement. The HBV-96 and XINANJIANG models are very different in process descriptions (especially in runoff generation) and parameters, although they share some common features and they both show robust model performances.

*We agree with your comment and added nuance to this statement by only referring to the number of model stores and the number of parameters. The text is revised: "The HBV-96 and XINANJIANG models most resemble each other regarding the number of stores and parameters; the parameters themselves and process descriptions differ. Both models contain a low number of instances, allowing the identification of the better-performing model.".*